

**Oxygen isotope composition of waters recorded in carbonates in strong clumped and oxygen**
**isotopic disequilibrium**
**Caroline Thaler[1*], Amandine Katz[1], Magali Bonifacie[1], Bénédicte Ménez[1], Magali Ader[1]**
[1] Université de Paris, Institut de physique du globe de Paris, CNRS, F-75005 Paris France
**\*corresponding author: Thaler.caroline@gmail.com**
**Abstract.** Paleoenvironmental reconstructions, which are mainly retrieved from oxygen isotope ($\delta^{18}O$)
and clumped isotope ($\Delta_{47}$) compositions of carbonate minerals, are compromised when carbonate
crystallization occurs in isotopic disequilibrium. To date, knowledge of these common isotopic
disequilibria, known as vital effects in biogenic carbonates, remains limited and the potential
information recorded by $\delta^{18}O$ and $\Delta_{47}$ offsets from isotopic equilibrium values is largely overlooked.
Additionally, in carbonates formed in isotopic equilibrium, the use of the carbonate $\delta^{18}O$ signature as a
paleothermometer relies on our knowledge of the paleowaters' $\delta^{18}O$ value, which is often assumed.
Here, we report the largest $\Delta_{47}$ offsets observed to date (as much as -0.270‰), measured on microbial
carbonates, that are strongly linked to carbonate $\delta^{18}O$ offsets (-25‰) from equilibrium. These offsets are
likely both related to the microorganism metabolic activity and yield identical erroneous temperature
reconstructions. Unexpectedly, we show that the $\delta^{18}O$ value of the water in which carbonates
precipitated, as well as the water-carbonate $\delta^{18}O$ fractionation dependence to temperature at equilibrium
can be retrieved from these paired $\delta^{18}O$ and $\Delta_{47}$ disequilibrium values measured in carbonates. The
possibility to retrieve the $\delta^{18}O$ value of paleowaters, sediments' interstitial waters or organisms' body
water at the carbonate precipitation loci, even from carbonates formed in isotopic disequilibrium, opens





long-awaited research avenues for both paleoenvironmental reconstructions and biomineralization
studies.

## 1    Introduction

Oxygen isotope composition ($\delta^{18}O$) paired with clumped isotope composition ($\Delta_{47}$) of carbonate
minerals is increasingly used for reconstructing paleoenvironmental or diagenetic conditions (Ghosh et
al., 2006; Mangenot et al., 2017; Henkes et al., 2018). The $\delta^{18}O$ composition of carbonates depends on
both the $\delta^{18}O$ value of the water in which the carbonate precipitated and the precipitation temperature
(Urey et al., 1951). Its use to reconstruct paleoenvironments can be combined with the new carbonate C-
O "clumped isotopes" abundancy ($\Delta_{47}$) thermometer which depends only on the carbonate precipitation
temperature (Ghosh et al., 2006). By combining the $\Delta_{47}$ derived temperatures and the carbonate $\delta^{18}O$
value ($\delta^{18}O_{carbonate}$), the $\delta^{18}O$ value of the water ($\delta^{18}O_{water}$) in which the carbonate precipitated can be
retrieved. However, this requires that solid carbonate and water reached isotopic equilibrium, which is
often hard to prove. Conversely, carbonate precipitation in isotopic disequilibrium is commonly
encountered (Affek et al., 2014; Loyd et al., 2016). Out of equilibrium $\delta^{18}O$ and $\Delta_{47}$ values are
particularly known to occur in biogenic carbonates (Thiagarajan et al., 2011, Bajnai et al., 2018)– the
most abundant carbonates in the sedimentary record. To date, the reasons for these isotopic disequilibria
in carbonates remain largely under-constrained. While $\Delta_{47}$ compositions of carbonates seemed at first
free of any biologically-driven or mineral-specific fractionation known to affect $\delta^{18}O_{carbonate}$
compositions (Eiler, 2011), recently identified disequilibrium $\Delta_{47}$ values (Saenger et al., 2012; Affek,
2013; Tang et al., 2014; Burgener et al., 2018) open new perspectives to unravel the mechanisms
responsible for oxygen isotopic disequilibrium in carbonate minerals. More specifically, it has become
crucial to determine if the $\delta^{18}O$ and $\Delta_{47}$ disequilibria observed in carbonates as diverse as those found in



coral reefs (Saenger et al., 2012), brachiopods (Bajnai et al., 2018), microbialites and methane seep
carbonates (Loyd et al., 2016), along with speleothems (Affek et al., 2014) could be explained by
oxygen-isotope disequilibria occurring in dissolved inorganic carbon (DIC) involved in carbonate
precipitation. In this case, $\delta^{18}O$ and $\Delta_{47}$ disequilibria in biogenic carbonates would record information,
however unavailable yet, on the physiological characteristics of carbonate-forming organisms.

In previous experiments we produced microbial calcium carbonates (Millo et al., 2012; Thaler et

al., 2017) that recorded the strongest oxygen isotope disequilibrium ever identified between DIC and
precipitation water (*i.e.* -25‰ offset from $\delta^{18}O_{carbonate}$ equilibrium values). We used carbonic anhydrase
(CA), an enzyme able to accelerate oxygen isotope equilibration between DIC and water *via* fast $CO_2$
hydration and $HCO_3^-$ dehydration. When CA was added to the precipitation water, the carbonate oxygen
isotope compositions reached equilibrium with the precipitation water (Thaler et al., 2017). Here, we
build up on these experiments as they offer a unique opportunity to assess experimentally whether
carbonates precipitated from DIC in disequilibrium with water also record $\Delta_{47}$ disequilibrium values, and
the type of information that is actually carried by these paired disequilibria. We latter show how and to
what extent this can be applied to previously published cases of oxygen isotopic offsets from
equilibrium values in both biogenic and abiotic carbonates.

**2    Material and Methods**
**2.1    Precipitation of microbial carbonates**
Carbonates were precipitated at 30±0.1°C using the procedure detailed in Millo et al., (2012) and Thaler
et al., (2017) and summarized hereafter. The precipitation solution (initial pH = 6.0) was composed of
ions added to Milli-Q® water (resistivity = 18 MΩ·cm) by dissolving salts in the following order:
$MgSO_4 \cdot 7H_2O$ (16 mM), NaCl (80 mM), KCl (4 mM), urea (33.3 mM), $CaCl_2$ (40 mM). The aim was to



mimic the ionic composition of a groundwater (Millo et al., 2012). In experiments with CA whose $\delta^{18}O$
results (but not the $\Delta_{47}$ ones) were recently published in Thaler et al., (2017), the precipitation solution
was supplemented with CA at a concentration of 2 mg/L. The precipitation solution (with or without
CA) was then mixed at a volumetric ratio of 1:1 with the ureolytic soil bacteria *Sporosarcina pasteurii*
suspended in Milli-Q® water, at a final optical density at 600 nm of 0.100±0.010. For this study, 16
gastight Exetainer® vials were filled with the precipitation solution without CA in order to sacrifice them
at regular time intervals (*i.e.* 30, 60, 120, 180, 360 min and 24 h) and thus obtain information on the
kinetics of the reaction, while reproducing the procedure followed for the experiment with CA (Thaler et
al., 2017) consisting of 27 vials sacrificed every 10 to 30 min and after 24 h. The vials capped with
rubber septa were filled up to the brim, *i.e.* without headspace, hence preventing any gaseous exchange
with the atmosphere or headspace gases. Ureolysis completion was followed by evaluating the
production of dissolved inorganic nitrogen (DIN = $NH_3+NH_4^+$). Determination of pH, DIN concentration
and amount of precipitated carbonates (Supplementary Fig. 2), as well as isotopic measurements, were
performed for each vial to monitor their evolution as the ureolysis reaction progresses. The pH initially
increased from 6.0 to 9.0 due to $NH_3$ production by ureolysis and consecutive alkalinization of the
precipitation solution (Supplementary Fig. 2a). The subsequent carbonate precipitation (Supplementary
Fig. 2b) lowered pH to 8.6 (without CA) and 8.5 (with CA) and was followed by a second pH increase
to 8.8 (without CA) and 8.7 (with CA) when carbonate precipitation stopped while ureolysis continued.
At ureolysis completion, all the calcium initially present in solution (*i.e.* the limiting reagent) has
precipitated whereas 35 to 45% of the DIC produced by ureolysis remained in solution. Carbonate
precipitates, formed at the bottom and on the wall of the vials, were immediately rinsed with a few drops
of pure ethanol in order to dehydrate bacteria and prevent further ureolysis, carbonate formation and/or
dissolution–reprecipitation processes. Ethanol was then removed, and prior to their collection,





carbonates were dried overnight at 40°C in the vials placed in a ventilated oven equipped with
desiccating beads. All of the measured chemical parameters (pH, DIC, amount of solid carbonates, $Ca^{2+}$
concentration, DIN) along with DIC and solid carbonate $\delta^{13}C$ behave similarly with or without active
CA (Thaler et al., 2017). It was not possible to measure $\Delta_{47}$ for all the precipitated carbonates due to
their low amount, particularly for the tubes sacrificed at the beginning of the experiments
(Supplementary Table 1).

## 99    2.2    $\delta^{18}O$ and $\Delta_{47}$ measurements and associated uncertainties

All the isotopic analyses were made at Institut de physique du globe de Paris (IPGP, France). $\delta^{18}O$
analyses were performed on carbonate powders of ca. 2 mg with a continuous helium-flow isotope ratio
mass spectrometer AP 2003 (Analytical Precision 2003, GV Instruments) coupled to a gas
chromatograph column (GC-IRMS, Chrompac Column Type 99960), as described in Millo et al., (2012)
and Thaler et al., (2017). External reproducibility on carbonate standards is ±0.1‰ (1SD) and represents
the uncertainty assigned to $\delta^{18}O_{carbonate}$ data.

The analytical procedure used for clumped isotope $\Delta_{47}$ measurements is only briefly presented

here and detailed in Bonifacie et al., (2017). About 5 mg of carbonates were digested at 90°C during 20
min with 104% phosphoric acid $H_3PO_4$ in a common acid bath. The produced gaseous $CO_2$ was purified
with a manual vacuum line before introduction into a Thermo Scientific MAT 253 dual-inlet mass
spectrometer. Each purified $CO_2$ gas was analyzed for their abundance in isotopologues with m/z from
44 to 49 versus a working gas provided by Oztech Trading Corporation with $\delta^{13}C$ = -3.71‰ VPDB and
$\delta^{18}O$ = +24.67‰ VSMOW, as determined with the international reference material NBS19. One single
$\Delta_{47}$ measurement corresponds to 70 cycles of 26 s integration time each (total integration time = 1820 s).
Conventional $\delta^{18}O$ and $\delta^{13}C$ data were also acquired simultaneously to $\Delta_{47}$ measurements with this



instrument (Supplementary Tables 1 and 3). They are in excellent consistency with data obtained with
the continuous-flow method on smaller samples (Supplementary Table 1).

The $\Delta_{47}$ is calculated as a function of the stochastic distribution of the $CO_2$ isotopologues, as

follows:
$$\Delta_{47} = \left[ \left( \frac{R^{47}_{measured}}{R^{47}_{stochastic}} - 1 \right) - \left( \frac{R^{46}_{measured}}{R^{46}_{stochastic}} - 1 \right) - \left( \frac{R^{45}_{measured}}{R^{45}_{stochastic}} - 1 \right) \right] \times 1000 \qquad (1),$$
where $\Delta_{47}$ is expressed in per mil (‰), and $R^{47}$, $R^{46}$ and $R^{45}$ are the abundance ratios of the masses 47,
46, 45 respectively, relative to the mass 44 ($^{12}C^{16}O^{16}O$). $R^{X}_{measured}$ are measured ratios inside the $CO_2$
sample. $R^{X}_{stochastic}$ are calculated from the measured 44, 45, 46 and 47 abundance ratios. The amount of
isotopologues of mass 47 (mainly $^{13}C^{18}O^{16}O$, but also $^{12}C^{17}O^{16}O$ and $^{13}C^{17}O^{17}O$) measured within the
$CO_2$ sample extracted from the acid digestion of the carbonates is linked to the amount of isotopologues
of mass 63 (mainly $^{13}C^{18}O^{16}O^{16}O$) within the reacted carbonate mineral (Guo et al., 2009). For the
correction from $^{17}O$ interferences we used the $^{17}O$ correction parameters from Brand et al., (2010), as
recently recommended (Daëron et al., 2016). In order to transfer the obtained raw $\Delta_{47}$ data into the
absolute Carbon Dioxide Equilibrated Scale "CDES" ($\Delta_{47\ CDES90}$ being the $\Delta_{47}$ values of carbonates
reacted within acid at 90°C), standards of $CO_2$ gases equilibrated at 25°C and 1000°C and with bulk
isotopic compositions covering the range of measured carbonate samples ($\delta^{47}$ values between -50 and
+24‰) were analyzed interspaced with unknown samples (typically 15 equilibrated $CO_2$ gas analyses by
discrete session of analysis, 4 analytical sessions in total; Supplementary Table 4). For each analytical
session, as recommended in Dennis et al., (2011), the $\Delta_{47}$ data were finally corrected with a fixed
Equilibrated Gas Line slope (only slightly varying from 0.0048 to 0.0062 over our analytical sessions)
and an Empirical Transfer Function (slopes varying from 1.0859 to 1.1344) based on the equilibrated
$CO_2$ standards. Finally, the accuracy of our whole dataset and processing procedure was validated on
carbonate reference material (*i.e.* IPGP-Carrara and 102-GC-AZ01), typically analyzed every 2



unknown samples (Supplementary Table 4). The $\Delta_{47}$ values obtained at IPGP over the course of this
study are $\Delta_{47\ CDES90} = 0.316\pm0.020‰$ (1SD, $n = 16$) for IPGP-Carrara and $\Delta_{47\ CDES90} = 0.620\pm0.010‰$
(1SD, $n = 18$) for 102-GC-AZ01. Those values are indistinguishable from the values obtained at IPGP
over four years of analyses on the same instrument ($n >300$) or previously reported by other laboratories
(Daëron et al., 2016).

**2.3      Temperature estimates and associated uncertainties**
Apparent temperatures issued from oxygen isotope compositions were calculated based on the measured
$\delta^{18}O_{carbonate}$ values of both the precipitated carbonate and the precipitation water in each experimental
vial (Supplementary Table 1) and using the equation of oxygen isotopes' fractionation between calcite
and water from Kim and O'Neil, (1997). Apparent temperatures issued from clumped isotope
compositions were calculated from $\Delta_{47\ CDES90}$ data using the composite universal $\Delta_{47}$-T calibration (Eq. 3
from Bonifacie et al., (2017) with T, the temperature). It is noteworthy that our main observations and
conclusions do not change if other calibrations to temperature are used for $\delta^{18}O$ and/or $\Delta_{47}$ (Kelson et al.,
2017) (see also Supplementary Discussion and Supplementary Table 2). For both proxies, the reported
uncertainties on temperature estimates correspond to the standard deviation of the mean of replicated
isotopic measurements of the same powder propagated in the calibration equation (but the actual errors
on the calibration themselves are not considered). Note that the long-term external reproducibility on
homogeneous calcite reference materials found in this study (*i.e.* ±0.020‰, 1SD) is used for samples
with only one measurement or with 1SD lower than 0.020‰ (Supplementary Tables 1 and 4,
Supplementary Discussion).

**3        Results and Discussion**



### 3.1    $\Delta_{47}$ and $\delta^{18}O$ compositions of microbial carbonates can present strongly correlated vital effects


We performed $\Delta_{47}$ measurements on (i) microbial carbonates precipitated without CA by faithfully
replicating the experiment detailed in Thaler et al., (2017) and (ii) microbial carbonates precipitated in
the presence of CA remaining from these experiments. These calcium carbonates were precipitated as
the result of microbially-driven hydrolysis of urea into DIC and ammonia (Millo et al., 2012). They
constitute a reliable model for carbonate precipitation triggered by enzymatic production or transport of
DIC, as it is the case for micro- and macro-skeletal carbonates common in the Phanerozoic, and for
microbially-mediated carbonates since the Precambrian.
Without CA, the isotopic values of the very first carbonate precipitates present strong isotopic offsets
from equilibrium values, down to -0.270‰ for $\Delta_{47}$ (the largest $\Delta_{47}$ offset ever measured in solid
carbonates) and -24.7‰ for $\delta^{18}O_{carbonate}$ (Fig. 1 and Supplementary Table 1). Both $\Delta_{47}$ and $\delta^{18}O_{carbonate}$
absolute values then increase as ureolysis progresses, reducing offsets from equilibrium values to -
0.179‰ for $\Delta_{47}$ and -15.7‰ for $\delta^{18}O_{carbonate}$. In the presence of CA, the trends observed for the $\Delta_{47}$ and
$\delta^{18}O_{carbonate}$ values are similar but the offsets from equilibrium are drastically reduced (down to -0.027‰
for $\Delta_{47}$ and -1.4‰ for $\delta^{18}O_{carbonate}$ at the end of the experiment; Fig. 1), hence attesting for on-going
isotopic equilibration of DIC with water by CA enzymatic activity prior to and during carbonate
precipitation. The comparable behavior of $\Delta_{47}$ and $\delta^{18}O_{carbonate}$ values with respect to CA suggests that
both disequilibria are inherited from the $\delta^{18}O$ and $\Delta_{47}$ signatures of the DIC generated by the biological
activity.

### 3.2    $\Delta_{47}$ and $\delta^{18}O_{carbonate}$ disequilibrium originate from the metabolic production of DIC






Here, we discuss the potential processes known to generate $\delta^{18}O$ and $\Delta_{47}$ isotope fractionations during
carbonate precipitation and we identify the main mechanism explaining our paired $\Delta_{47}$ and $\delta^{18}O_{carbonate}$
disequilibria. The relatively high precipitation rate (R) in our experiments (log R = -3.95 mol· m$^{-2}$· s$^{-1}$;
(Thaler et al., 2017)) can only account for an oxygen kinetic isotope fractionation (KIF) of about 1 to
2‰ for $\delta^{18}O$ values (Watkins et al., 2013), while the oxygen isotope disequilibrium recorded in our
carbonates reaches -24.7‰. Degassing of $CO_2$, known to fractionate DIC oxygen isotopes (Affek and
Zaarur, 2014), can be ruled out as there is no gas phase in our experiments (see Material and Methods).
Any potential kinetic fractionation due to DIC diffusion (Thiagarajan et al., 2011) is also unlikely as
precipitation occurred on the bacterial DIC-producing cells, as highlighted by scanning electron
microscopy showing bacterial cells trapped within and at the surface of carbonate crystals
(Supplementary Fig. 1). Accordingly, the large offsets from equilibrium values observed for both $\Delta_{47}$
and $\delta^{18}O$ in our microbial carbonates can only result from (i) a KIF induced by $CO_2$
hydration/hydroxylation into $HCO_3^-$ (but only if ureolysis produces $CO_2$ rather than $H_2CO_3$, which has
not been established yet (Matsuzaki et al., 2013)) or (ii) a metabolic isotopic signature of the DIC
produced by the bacteria, inherited from the initial isotopic composition of urea and/or due to a KIF
introduced by the urease enzyme. $CO_2$ hydration/hydroxylation leads to the formation of $HCO_3^-$, with
two oxygen atoms coming from $CO_2$ and the third one from $H_2O$ (hydration) or $OH^-$ (hydroxylation).
The $\delta^{18}O_{HCO3^-}$ value can then be estimated using a simple mass balance calculation (Létolle et al., 1990b;
Usdowski et al., 1991). The newly formed $HCO_3^-$ is enriched in $^{16}O$ compared to the reacting $CO_2$
because of the incorporation of oxygen coming from $H_2O$ or $OH^-$, both enriched in $^{16}O$ relative to $^{18}O$ in
contrast with $CO_2$ (Green and Taube, 1963;Beck et al., 2005). Such a low $\delta^{18}O_{HCO3^-}$ value, several per
mil lower than the equilibrium one, can then be preserved in the calcium carbonate if precipitation
occurs shortly after $CO_2$ hydration/hydroxylation and before the full equilibration with water (Rollion-





Bard et al., 2003). Regarding clumped isotopes, ab initio calculations predict that the fractionation
associated with $CO_2$ hydration/hydroxylation increases the relative abundance of $^{13}C$-$^{18}O$ bonds, and
thus the $\Delta_{47}$ value (Guo, Ms 2009). Even though this predicted fractionation trend has previously been
used to explain several datasets for which $CO_2$ hydroxylation was assumed to occur prior to carbonate
precipitation (Tripati et al., 2015; Spooner et al., 2016), such a tendency can only be validated using data
acquired on carbonates for which $CO_2$ hydration/hydroxylation is demonstrated. This is the case of (i)
hyperalkaline travertines (Falk et al., 2016) even though part of the reported kinetic isotope fractionation
can be interpreted as resulting from $CO_2$ dissolution process (Clark et al., 1992) and (ii) two
experimental samples (Tang et al., 2014) precipitated at high pH where $CO_2$ hydroxylation dominates.
Both studies show, in agreement with the ab initio calculations (Guo, Ms 2009), higher $\Delta_{47}$ and lower
$\delta^{18}O_{carbonate}$ values compared to equilibrium. Thus, in a case where ureolysis would produce $CO_2$ in
isotopic equilibrium with water, the $\Delta_{47}$ values affected by $CO_2$ hydration/hydroxylation recorded in
calcium carbonates should be higher than the equilibrium value, while our microbial carbonates are
showing $\Delta_{47}$ values lower than equilibrium. Thus, we conclude that our low $\Delta_{47}$ values measured in
carbonates can only be explained by a metabolic source effect. In our case it corresponds to the ureolytic
production of DIC, either directly as $H_2CO_3$ or as $CO_2$, with a $\Delta_{47}$ value low enough to compensate for
any potentially succeeding increase due to the KIF associated with $CO_2$ hydration/hydroxylation.
Nonetheless, the slow but continuous increase observed in our experiment without CA for both $\Delta_{47}$ and
$\delta^{18}O_{carbonate}$ values more likely reflects ongoing equilibration of DIC oxygen isotopes with water at a
slow rate.

Our results highlight that the isotope clumping proceeds continuously as C-O bonds are breaking

and re-forming in the DIC, allowing oxygen isotopes ($^{16}O$, $^{17}O$ and $^{18}O$) to be redistributed between
$H_2O$, $OH^-$, $H_2CO_3$, $HCO_3^-$ and $CO_3^{2-}$ species *via* $H_2O/OH^-$ -attachment to $CO_2$ and -detachment from



HCO$_3^-$. In the experiment with CA, both $\Delta_{47}$ and $\delta^{18}O_{carbonate}$ reach simultaneously values close to
equilibrium and without CA both $\Delta_{47}$ and $\delta^{18}O_{carbonate}$ values increase simultaneously. This coevolution
corroborates former observations of comparable kinetics for clumped isotopes and $\delta^{18}O$ equilibration
between DIC and water or $CO_2$ and water once $\delta^{13}C$ is equilibrated (Affek, 2013; Clog et al., 2015). This
principle has been used to correct for disequilibrium fractionation factor in speleothems (Affek et al.,

2008).


**3.3    Erroneous yet comparable temperatures reconstructed from disequilibrium $\Delta_{47}$ and $\delta^{18}O$**
**values in carbonates**
Apparent temperatures were calculated from disequilibrium $\Delta_{47}$ values obtained in the experiment
without CA using calibration of Bonifacie et al., (2017). Ranging from 198±21°C to 115±8°C (Fig. 2),
they are at odds with the actual precipitation temperature of 30±1°C (see Methods). This shows that
when carbonates precipitate from DIC in oxygen-isotope disequilibrium with water, the abundance of
$^{13}C–^{18}O$ bonds in carbonates does not correlate with precipitation water temperature. Conversely, the
temperatures reconstructed from the $\Delta_{47}$ values of carbonates precipitated in the presence of CA, ranging
from 47±6°C to 39±2°C, are much closer to the actual precipitation temperature. Interestingly, the
apparent temperatures reconstructed using Kim and O'Neil et al., (1997) calibration from the $\delta^{18}O_{carbonate}$
and $\delta^{18}O_{water}$ values of the same samples show comparable offsets from the actual temperature in both
experiments without CA (from 218±2°C to 139±1°C) and with CA (from 39±1°C to 37±1°C) (Fig. 2).
Practically, this implies that similar temperatures calculated from both carbonate $\Delta_{47}$ and $\delta^{18}O_{carbonate}$
values (in a case where the precipitation water $\delta^{18}O$ can be determined) can neither constitute evidence
against O-isotope disequilibrium nor confirm that this is the true precipitation temperature.





**3.4    $\Delta_{47}$ and $\delta^{18}O_{carbonate}$ paired disequilibria record the $\delta^{18}O$ of the water in which the**
**carbonates precipitated**
The fact that both $\Delta_{47}$ and $\delta^{18}O_{carbonate}$ values permit to calculate similarly evolving apparent
temperatures along the (dis)equilibration profile recorded in carbonates as the experiment proceeds,
indicates that the $\delta^{18}O_{carbonate}$, $\delta^{18}O_{water}$, $\Delta_{47}$, and apparent temperature values are all together linked. In a
$\Delta_{47}$ versus $\delta^{18}O_{carbonate}$ diagram, all of our data align, irrespectively of the fact that they are in strong
isotopic disequilibrium or close to equilibrium (Fig. 3). Their alignment is fitted with what would be
expected for equilibrium $\Delta_{47}$ and $\delta^{18}O_{carbonate}$ values of calcite precipitated at various temperatures from a
water at a given $\delta^{18}O_{water}$ value. This $\delta^{18}O_{water}$ value can be calculated by combining for the same
temperature, the equations of $\Delta_{47}$ and $\delta^{18}O_{carbonate}$ temperature calibrations from Kim and O'Neil, (1997)
and Bonifacie et al., (2017), respectively (Eq. 2):
$$\delta^{18}O_{water} = \exp\left|-\frac{18.03}{\sqrt{\frac{0.0422\times10^{6}}{\Delta_{47\,CDES90}-0.1126}}} + 32.42 \times 10^{-3} + \ln(\delta^{18}O_{carbonate} + 1000)\right| - 1000 \quad (2),$$
with $\delta^{18}O_{water}$ and $\delta^{18}O_{carbonate}$ values in the same isotopic referential (here VSMOW), and $\Delta_{47}$ values
reported into the absolute Carbon Dioxide Equilibrated Scale ($\Delta_{47\,CDES90}$). The calibration of Kim and
O'Neil, (1997) was preferred over more recent calibration equations (e.g. Watkins et al., 2013) because
it provides the best consistency for temperatures reconstructed from both the carbonate $\delta^{18}O$ and $\Delta_{47}$
values at temperatures above 100°C. Note that Kim and O'Neil, (1997) and Bonifacie et al., (2017)
calibrations were developed independently, which prevents circular reasoning. Finally, as Kim and
O'Neil, (1997) is the most used calcite calibration to date, it also allows for a broader comparison with
previously published results.
Despite the fact that the data present a large range of offsets from equilibrium (Fig. 3), the mean
$\delta^{18}O_{water}$ value calculated using Eq. 2 for each combination of $\Delta_{47}$ and $\delta^{18}O_{carbonate}$ values measured for
our carbonates is -8.0±2.8‰ (1SD), indistinguishable (*i.e.* within errors) from the $\delta^{18}O_{water}$ values



measured in our experiments (-6.4±0.2‰ with CA and -6.8±0.2‰ without CA; Fig. 3). Note that such
precision in $\delta^{18}O_{water}$ values found in disequilibrium carbonates is remarkable considering that even for
equilibrium carbonates, $\delta^{18}O_{water}$ can only be retrieved from paired $\Delta_{47}$ and $\delta^{18}O_{carbonate}$ values with a
precision of ±1‰ at best (see Supplementary information). This opens the promising opportunity to
retrieve the $\delta^{18}O$ value of the water in which carbonates precipitated out of equilibrium for both
$\delta^{18}O_{carbonate}$ and $\Delta_{47}$.

In order to evaluate the applicability of such an approach to other types of carbonates, Fig. 4

compiles disequilibrium paired $\delta^{18}O_{carbonate}$ and $\Delta_{47}$ data from two previously published experimental
studies (Tang et al., 2014; Staudigel et al., 2018). These studies were chosen to further evaluate the
relevancy of our $\delta^{18}O_{carbonate}$- $\Delta_{47}$ correlation because they are the only published dataset reporting full
sets of _measured_ $\delta^{18}O_{water}$, $\delta^{18}O_{carbonate}$ and $\Delta_{47}$ values, together with precipitation temperatures. A
perfect knowledge (_i.e._ measurements and not estimates) of these four parameters is mandatory here to
adequately test whether the use of our new $\delta^{18}O_{water}$ proxy could be generalized to a large diversity of
carbonates. This thus precludes plotting in Fig.4 most published $\Delta_{47}$ studies on both natural and
experimental samples, in which $\delta^{18}O_{water}$ and/or temperature were not directly measured. These two
datasets are also recent enough to allow the conversion of their $\Delta_{47}$ values to the currently used
normalization method (_i.e._ the CDES absolute reference frame). It will then allow comparison with
future studies, if measuring and reporting all these four parameters together become the rule rather than
the exception in $\Delta_{47}$ studies. Fig. 4a shows paired $\delta^{18}O_{carbonate}$ and $\Delta_{47}$ values of abiotic carbonates
produced at 5, 25 and 40°C that are known to be affected by KIF due to fast precipitation and for at least
two of them by KIF due to $CO_2$ hydration/hydroxylation prior to precipitation (Tang et al., 2004). Except
for these two carbonate samples, the data align on a $\Delta_{47}$ versus $\delta^{18}O_{carbonate}$ covariation curve that cannot
be explained by sole temperature variation. As for our microbial carbonates obtained with or without



CA, the average calculated $\delta^{18}O_{water}$ (Eq. 2; -11.2±1.5‰) matches within error with the measured
$\delta^{18}O_{water}$ (-9.6±0.2‰) (Dietzel et al., 2009).

Fig. 4b shows paired $\delta^{18}O_{carbonate}$ and $\Delta_{47}$ values of abiotic carbonates that were precipitated

during an initial $CO_2$ degassing + equilibration phase followed by solely equilibration with water at 5,
15 and 25°C (Staudigel et al., 2018). During the latter equilibration phase, even though the carbonates
precipitated out of isotopic equilibrium, the paired $\delta^{18}O_{carbonate}$ and $\Delta_{47}$ values align on a covariation
curve of average calculated $\delta^{18}O_{water}$ value  (Eq. 2; -3.0±1.1‰) close to the measured $\delta^{18}O_{water}$ (-0.65‰).
As a major outcome of this study, we thus anticipate that reliable $\delta^{18}O_{water}$ values of precipitation water
can be retrieved from carbonates presenting $\Delta_{47}$ and $\delta^{18}O_{carbonate}$ values in strong disequilibrium.

Some data presented in Fig. 4 also permit to evaluate the conditions of applicability of our

approach. In Fig. 4a, the two data points deviating from the $\Delta_{47}$ versus $\delta^{18}O_{carbonate}$ covariation curve
correspond to carbonates precipitated at pH ~10 and 5°C (while the others formed at pH and
temperatures ranging from 8.3 to 9 and 5 to 40°C, respectively) that have recorded a KIF due to $CO_2$
hydration/hydroxylation prior to precipitation (Tang et al., 2014). At pH=10, $CO_2$ reacts at 95% with
$OH^-$ and at 5°C, DIC isotopic equilibration with water takes days. To a lesser extent, the KIF induced by
$CO_2$ hydroxylation seems also visible at pH= 9 (and 40°C) where $CO_2$ reacts at 82% with $OH^-$ but DIC
isotopic equilibration with water at 40°C only takes about 15 hours. As previously detailed, the direction
of these isotopic offsets from equilibrium is compatible with ab initio calculations (Guo, Ms, 2009) and
can be intuitively understood as follows: in carbonates derived from $CO_2$ hydroxylation, the $R^X_{stochastic}$
term used for the $\Delta_{47}$ calculation (Eq. 1) should be strongly modified as the $^{18}O$ concentration in $OH^-$ and
$H_2O$ is lower than in $CO_2$ and the reaction does not add more $^{13}C$ than what is present in $CO_2$. This
might explain why in the case of disequilibria acquired through $CO_2$ hydroxylation, the correlation
between paired $\delta^{18}O$ and $\Delta_{47}$ disequilibria and the precipitation water $\delta^{18}O$ is not preserved and $\delta^{18}O_{water}$

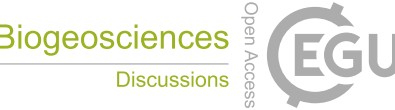

cannot be reconstructed by the approach proposed here. The negative slope associated with this KIF on
the $\Delta_{47}$ and $\delta^{18}O_{carbonate}$ diagram (Fig. 4a) is nevertheless a good tool to identify $CO_2$ hydroxylation
reactions.

In Fig. 4b, during the $CO_2$ degassing phase of the precipitation experiment (Staudigel et al.,

2018), the data also deviate from the $\delta^{18}O$ versus $\Delta_{47}$ covariation curve. This behavior was interpreted by
the authors as a decoupling between $\Delta_{47}$ and $\delta^{18}O_{carbonate}$ values due to variable kinetics of $^{12}C$-O and
$^{13}C$-O bounding. A known difference in equilibration kinetics takes place between C and O isotopes in
the carbonate system as carbon isotopes equilibrate in seconds, while oxygen isotopes necessitate
minutes to hour to equilibrate among the different oxygen-bearing species (*i.e.* $CO_2$, $HCO_3^-$, $CO_3^{2-}$, $H_2O$,
$OH^-$), depending on the pH, temperature and salinity of the solution (Zeebe and Wolf-Gladrow, 2001).
However, note that in that experiment, the carbon isotope compositions evolved for several hours as a
result of $CO_2$ degassing (Staudigel et al., 2018). We propose here that $CO_2$ degassing, because it affects
both C and O isotopes, modifies the $R^X_{stochastic}$ term (in Eq. 1), thus preventing $\Delta_{47}$ and $\delta^{18}O$ to vary with
the proportionality that allows to retrieve the $\delta^{18}O_{water}$ value on a $\Delta_{47}$ versus $\delta^{18}O$ covariation plot.
Hence, as for $CO_2$ hydroxylation, in case of a KIF induced by $CO_2$ degassing, $\delta^{18}O_{water}$ cannot be
reconstructed exclusively from disequilibrium $\delta^{18}O_{carbonate}$ and $\Delta_{47}$ values.

In summary, we conclude that mechanisms that can drastically change the $R^X_{stochastic}$ term in $\Delta_{47}$

calculation (such as $CO_2$ hydroxylation and degassing) prevent $\delta^{18}O_{water}$ reconstructions from paired
disequilibrium $\Delta_{47}$ and $\delta^{18}O_{carbonate}$ values.  Nevertheless, these mechanisms lead to peculiar types of
carbonates (*i.e.* speleothems that form in caves for $CO_2$ degassing, and travertine that form on lands
where fluids and gas escape from subsurface reservoirs for $CO_2$ hydroxylation) that represent only a
small fraction of all the carbonates existing on Earth. We hypothesize that ureolysis, which consists in
two successive steps of urea hydrolysis, an exchange reaction with the $H_2O$ molecule from the aqueous



medium, might give a DIC whose $R^X_{stochastic}$ term in $\Delta_{47}$ calculation is already close to that of a DIC
under equilibration with the $\delta^{18}O_{water}$. This would explain why even our most extreme out of equilibrium
carbonates still fall close to the $\Delta_{47}$ versus $\delta^{18}O_{carbonate}$ covariation line corresponding to the real $\delta^{18}O_{water}$
value.

### 3.5    Toward a better understanding of body water $\delta^{18}O$ in biomineralizing organisms

The ability to reconstruct precipitation water $\delta^{18}O_{water}$ from disequilibrium $\Delta_{47}$ and $\delta^{18}O_{carbonate}$ values
further allows to examine the origin of the vital effect observed in organisms for which (i) $CO_2$
degassing and hydration/hydroxylation KIF can be ruled out, and (ii) only small $\delta^{13}C$ variations are
observed, thus preserving the $R^X_{stochastic}$ term in $\Delta_{47}$ calculation. We hypothesize that such an approach
could open perspectives to understand how $\Delta_{47}$ and $\delta^{18}O$ signals are affected by kinetic effects in most of
the biogenic carbonates, provided that $CO_2$ hydroxylation or degassing do not occur prior to carbonate
precipitation. This approach could thus be applied to the vast majority of sedimentary carbonates
(Milliman et al., 1993) and since deep time (*i.e.* microbialites, brachiopods, bryozoans, bivalves,
foraminifera, coccoliths), even when $\delta^{18}O_{carbonate}$ variations occur in the shell of the organism.
Additionally, the data presented here stand as an experimental demonstration that the mechanisms
controlling carbonate $\delta^{18}O$ equilibration with water (*i.e.* DIC equilibration with water) also control solid
carbonate $\Delta_{47}$ equilibrium (Watkins et al., 2015). This result can be used to recover information on
biomineralization mechanisms. For example, in recent coccolithophorid *Emiliana huxleyi* culture
experiments, the calcitic shell produced by the organism systematically yields a 2‰ positive $\delta^{18}O$ offset
from equilibrium values while their $\Delta_{47}$ values seem to faithfully record precipitation temperature (Katz
et al., 2017). These coccolithophorids were grown in waters with different $\delta^{18}O_{water}$ compositions (*i.e.*
measured at -6.14, -5.82 and 0.65‰ VSMOW, that are respectively seawater A, B and C in Fig. 4c).



Based on our results, which demonstrate that no $\delta^{18}O$ disequilibrium should be recorded in solid
carbonates if the associated $\Delta_{47}$ is at equilibrium, we can assume that the coccoliths precipitated at
oxygen-isotope equilibrium and calculate from Eq. 2 the actual $\delta^{18}O$ value of the water in which
precipitation took place (respectively shifted by 1.0±0.2‰, 2.1±0.4‰ and 1.1±0.7‰ towards more
positive values compared to the $\delta^{18}O_{water}$ value measured for the culture medium water; Fig. 4c). This
could reflect a biologically-driven difference between the $\delta^{18}O$ of body water at the precipitation site
inside *E. huxleyi* and the $\delta^{18}O$ of ambient water (*i.e.* the culture medium water). This hypothesis is
supported by what is known about intracellular precipitation of coccolith performed by
coccolithophorids: each coccolith forms from the accumulation of coccolithosomes, which are vesicles
containing up to a dozen of 7 nm spherical calcium-rich granular units (Outka and Williams, 1971).
Water in these ~100 nm vesicles can be considered as a finite reservoir whose isotopic composition
could be modified through isotopic exchange with a DIC affected by metabolic isotope fractionation.
Another mechanism that could increase the $\delta^{18}O$ value of a finite water reservoir by equilibrating it with
a comparable reservoir of DIC would be the introduction of DIC systematically as $CO_2$. As $HCO_3^-$ and
$CO_3^{2-}$ are enriched in $^{16}O$ in comparison to $CO_2$, the $CO_2$ conversion to $HCO_3^-$ and $CO_3^{2-}$ at equilibrium
before precipitation would pump $^{16}O$ from water.

In both case scenarios, a local change in water isotopic composition requires that the water

molecules turnover (*i.e.* external inputs) in these cellular organites is slow enough. Coccolithosomes are
subunits of the Golgi complex, which is a system of flat stacked vesicles concentrating a lot of
membranes in a small location (Outka and Williams, 1971). It is thus plausible that in a single celled
organism, living in seawater and performing intracellular biomineralization, specific osmolarity and
water circulation regulation mechanism are occurring. It is particularly plausible in the Golgi complex,
whose water content is isolated from seawater by several membranes. We thus suggest that inside



coccolithosomes, coccoliths precursors precipitate in equilibrium with the body water for oxygen
isotopes, but that the body water has a different $\delta^{18}$O value than the seawater, which explains the
observed $\delta^{18}$O apparent fractionation while their $\Delta_{47}$ composition reflects culture temperature (Katz et
al., 2017). It has already been highlighted through geochemical analysis of coccoliths, that
coccolithosomes water has altered pH (Liu et al., 2018) and ion concentrations (Hermoso et al., 2017) in
comparison to seawater. We hypothesize that the internal $\delta^{18}$O water would thus be another parameter
controlled by the coccolithophore algae.

**3.6     Ubiquity of the observed $\delta^{18}$O$_{carbonate}$- $\delta^{18}$O$_{water}$-$\Delta_{47}$-temperature covariations in both**
**equilibrium *and* disequilibrium carbonates**
As shown above, in a $\Delta_{47}$ versus $\delta^{18}$O$_{carbonate}$ diagram, disequilibrium carbonates precipitated at fixed
temperature plot on the theoretical line of equilibrium carbonates precipitated with a similar $\delta^{18}$O$_{water}$ but
at a different temperature. This is illustrated in Fig. 5 where the three disequilibrium data series studied
in this paper (Fig. 3 for this study and Figs. 4a and b for datasets from Tang et al., (2014) and Staudigel
et al., (2018)) align with equilibrium data series. In other words, the values of the disequilibrium
1000ln$\alpha_{carbonate\text{-}water}$ for oxygen isotopes (with $\alpha = \frac{\delta^{18}O_{carbonate}+1000}{\delta^{18}O_{water}+1000}$) are similar to the equilibrium
1000ln$\alpha_{carbonate\text{-}water}$ for any given, and independently determined, apparent $\Delta_{47}$ temperature (Fig. 5). In
details, our closest to equilibrium data recording low apparent temperatures match better the predicted
equations from Coplen, (2007) and Watkins et al.,(2013), recently updated (Daëron et al., 2019). This
latter calibration is based on carbonates from two caves where calcite precipitate extremely slowly and is
thus assumed to have precipitated at equilibrium. Note that the use of these two cave samples for
determining the dependence to temperature of the equilibrium 1000ln$\alpha_{carbonate\text{-}water}$ relies on the



assumption that constant environmental conditions, including temperature in the two caves (7.9 and
33.7°C) and the $\delta^{18}O_{water}$ value of the precipitation water, prevailed over the whole period of carbonate
precipitation (Coplen, 2007; Kluge et al., 2014). In Fig. 5, the disequilibrium data recording high
apparent temperatures (above 100°C) match better the predicted equation of Kim and O'Neil, (1997).
This $1000ln\alpha_{carbonate\text{-}water}$ dependence to temperature was established on carbonates precipitated in the
laboratory at well-known $\delta^{18}O_{water}$ and temperatures (from 10 to 40°C), but suspected to present a small
KIF due to a high precipitation rate that lowers the value of the $1000ln\alpha_{carbonate\text{-}water}$ (Watkins et al.,
2013). Despite this, we used this equation to retrieve the $\delta^{18}O_{water}$ from our experimental carbonates,
because most of them are associated with high apparent $\Delta_{47}$ temperatures. Coplen, (2007) or Watkins et
al., (2013) equations would have return 2‰ lower values (ca. -10±2‰ compared to -8±3‰ calculated
with Kim and O'Neil (1997) equation). This shows how crucial it is to improve knowledge on the
equilibrium $1000ln\alpha_{carbonate\text{-}water}$ at high temperatures in order to improve the accuracy and precision of
our new proxy for reconstructing the $\delta^{18}O_{water}$ from which carbonates, even disequilibrium ones,
precipitated.

Importantly, we here establish a new method to determine the equilibrium $1000ln\alpha_{carbonate\text{-}water}$,

which consists in using the kinetics of $\Delta_{47}$ and $\delta^{18}O$ covariations during (dis)equilibration. Notably,
because of the very large range of apparent temperatures recorded by disequilibrium carbonates
(between ~40 and 200°C; Fig. 5) this method could be particularly adapted to calibrate $1000ln\alpha_{carbonate\text{-}}$
$_{water}$ at high temperatures for which the differences between the two most popular $1000ln\alpha_{carbonate\text{-}water}$
dependence to temperature equations (Kim and O'Neil, 1997; Coplen, 2007) appear larger (Fig. 5).
Unfortunately, none of the three experimental setups having produced these disequilibrium carbonates
(this study, as well as Tang et al., 2014 and Staudigel et al., 2018) were designed for the purpose of
calibrating the equilibrium $1000ln\alpha_{carbonate\text{-}water}$. It is thus not possible using these datasets to propose a



meaningful calibration. At least in our experiment, too many phenomena including the relatively high
precipitation rate, variations in $\delta^{13}C$ values (~3‰) (Thaler et al., 2017), and the presence of traces of
aragonite and vaterite in our carbonates (Supplementary Information) lower the accuracy of the
reconstructed equilibrium $1000ln\alpha_{carbonate-water}$ values.

As a broader perspective, we anticipate that such an approach will help in determining critical

equilibrium fractionation factors for other gaseous isotopic systems (such as isotopologues of molecules
containing S-O bounds) or minerals of prime interest in biology and geology if clumped isotopes
measurements expand further beyond gaseous mass spectrometry (*e.g.* bounding between Fe-O, Fe-S,
Ca-C).

**4.    Conclusions**
Our experimental results show that the information held in disequilibrium (and apparent disequilibrium)
carbonates is diverse and promising. First, a paired $\Delta_{47}$ and $\delta^{18}O_{carbonate}$ disequilibrium indicates that
carbonates have precipitated in a dynamic environment where DIC and water did not reach isotopic
equilibrium. In our microbial carbonate experiments, all the DIC is produced in isotopic disequilibrium
with water and precipitates rapidly. Accordingly, the disequilibrium O isotope compositions recorded in
those carbonates are maximized compared to what can be expected in nature where newly produced DIC
is expected to be mixed with at least partly equilibrated ambient DIC before carbonates precipitate.
Second, the combined use of clumped and traditional oxygen isotopic compositions allows retrieving the
$\delta^{18}O$ of the precipitation water, *i.e.* organism body water or environmental water, even for carbonates
presenting $\delta^{18}O$ and/or $\Delta_{47}$ disequilibria or apparent disequilibria. Hence, except in the case of processes
such as $CO_2$ degassing and $CO_2$ hydration/hydroxylation, which likely modify the $R^X_{stochastic}$ term in $\Delta_{47}$
calculation, paired $\Delta_{47}$ and $\delta^{18}O_{carbonates}$ disequilibria in carbonates can be used to reconstruct the

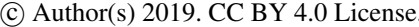



oxygen-isotope composition of both DIC and water at the precipitation loci even when precipitation
occurred under disequilibrium conditions. Third, the (dis)equilibration trend in a $\Delta_{47}$ versus $\delta^{18}O_{carbonates}$
covariation diagram can be used as a new method to determine the equilibrium fractionation factor
between carbonate and water for a wide range of temperatures. Altogether, this open up new avenues to
better constrain not only past climate changes through improved paleoenvironmental reconstructions but
also the physiology and habitat of sea-life sensitive to ocean acidification.

**Data availability.** All the data generated and analyzed in this study are available within the paper and in
its Supplementary Information.

**Author contributions**
C.T. and A.K. conceived the research. C.T. performed the microbial precipitation experiment and the
$\delta^{13}C$ and $\delta^{18}O$ analyses during her PhD thesis under M.A. and B.M. supervision. A.K. performed the $\Delta_{47}$
analyses during her PhD thesis under M.B. supervision. C.T. took the lead in the interpretation of the
results and the writing of the original draft. All authors provided critical feedback and helped shaping
the research, analyses and manuscript.

**Competing interests** The authors declare no competing financial interests.

**Acknowledgements**
This research was supported by French MRT PhD fellowships to C.T. and A.K., the Centre de
Recherches sur le Stockage Géologique du CO$_2$ (IPGP-TOTAL-Schlumberger-ADEME) (B.M. and





M.A.) and an Emergence grant from the Paris council to M.B. This study contributes to the IdEx
Université de Paris ANR-18-IDEX-0001.

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

**Figures & Figure Legends**



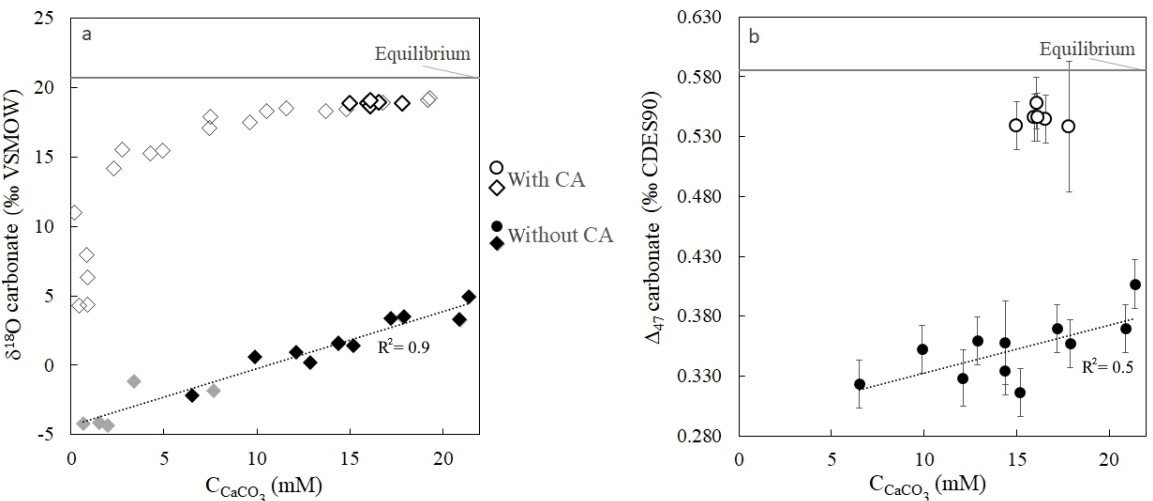


**Figure 1 | Strong $\delta^{18}O$ and $\Delta_{47}$ disequilibria recorded in microbial carbonates** as shown by $\delta^{18}O_{carbonate}$ (**a**) and $\Delta_{47}$ (**b**) values of calcium carbonates ($CaCO_3$) precipitated during bacterial ureolysis at 30°C (with and without carbonic anhydrase, CA; open and solid symbols, respectively) as a function of carbonate accumulation ($C_{CaCO3}$). Black symbols correspond to samples for which both $\Delta_{47}$ and $\delta^{18}O$ measurements were performed. The grey horizontal lines are equilibrium $\delta^{18}O_{carbonate}$ and $\Delta_{47}$ values at 30°C for calcite following Bonifacie et al., (2017) and Kim and O'Neil, (1997) calibrations, respectively. Uncertainties (one standard deviation, 1SD) are smaller than symbol for $\delta^{18}O$ and $C_{CaCO3}$ values (Supplementary Table 1). Reported $\Delta_{47}$ uncertainties are detailed in Methods and Supplementary Discussion.

647



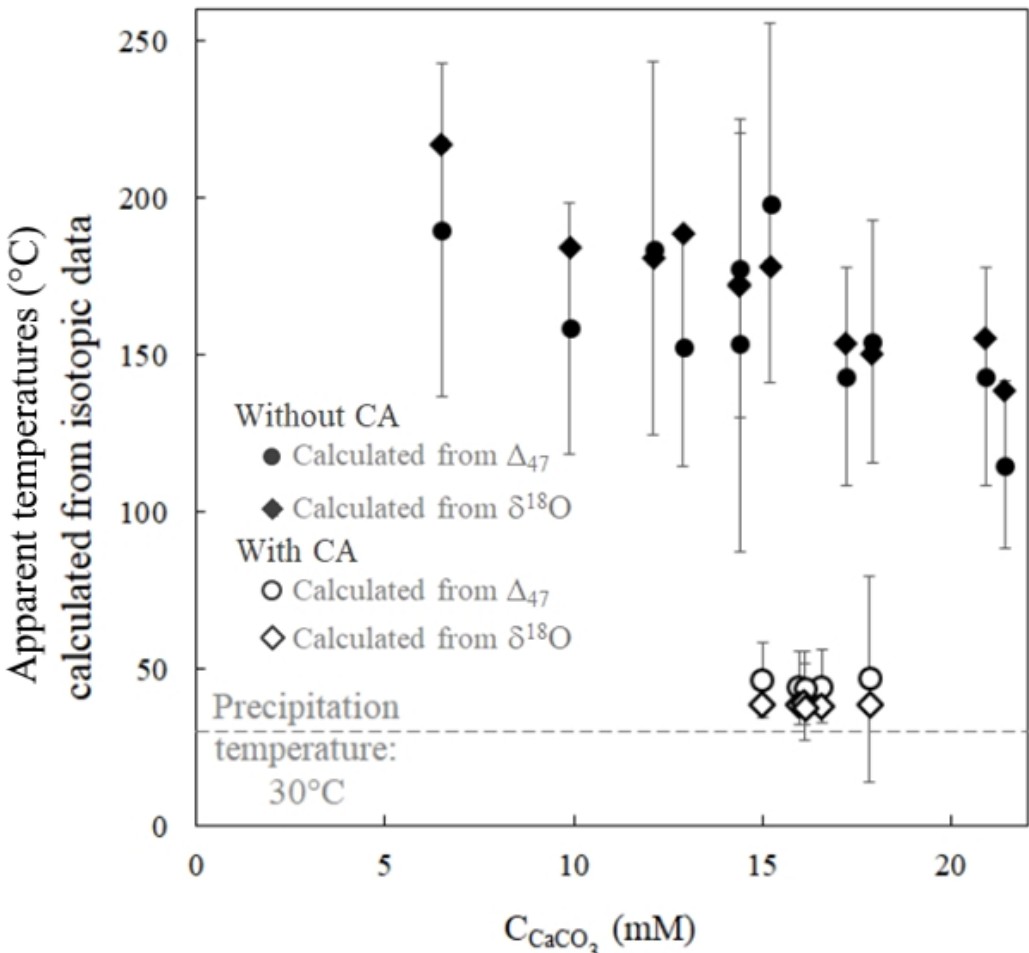

**Figure 2 | $\delta^{18}O_{carbonate}$ and $\Delta_{47}$ disequilibria in microbial carbonates induce comparable biased estimates of precipitation temperature** as illustrated by apparent temperatures calculated from the carbonate $\delta^{18}O_{carbonate}$ and $\Delta_{47}$ signatures as a function of $CaCO_3$ accumulation. Open and solid symbols refer to the experiments with and without CA, respectively. The dashed grey line corresponds to the actual precipitation temperature. Apparent temperatures are respectively calculated from the $\delta^{18}O_{carbonate}$ and $\Delta_{47}$ calibrations to temperature of Bonifacie et al., (2017) and Kim and O'Neil, (1997). Reported uncertainties were calculated as the propagation of the one standard deviation (1SD) error of the isotopic data in the calibration equations (Supplementary Information).

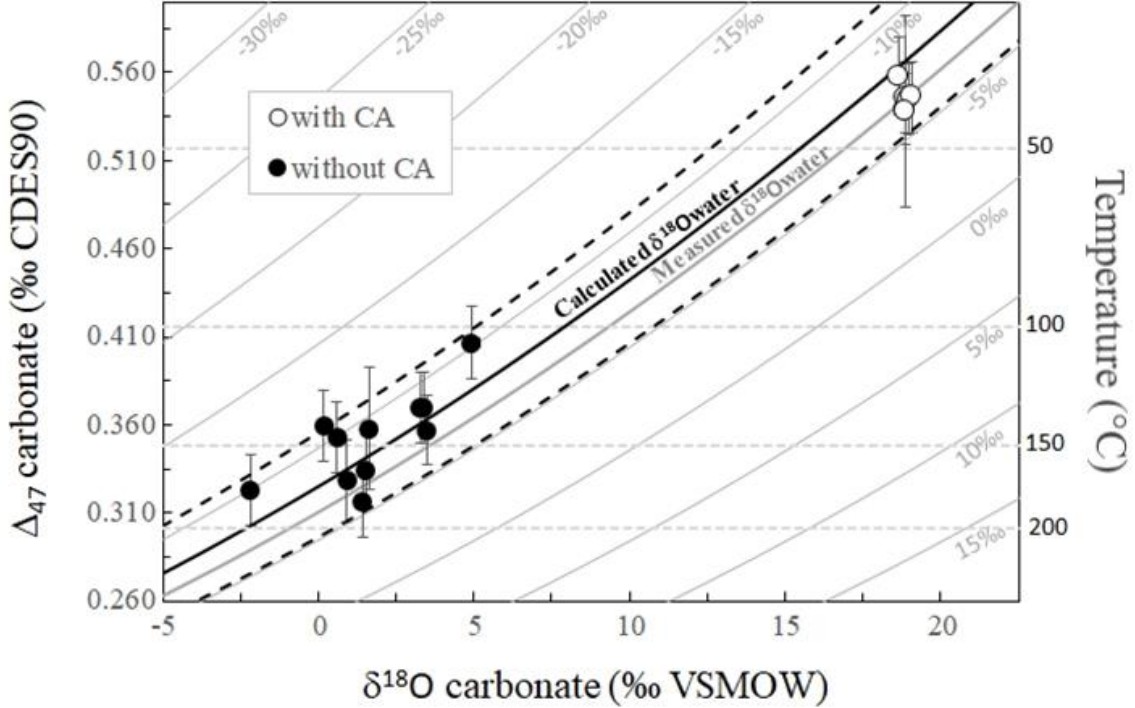


**Figure 3 | Combined $\delta^{18}O_{carbonate}$ and $\Delta_{47}$ disequilibria of microbial carbonates precipitated at**
**30°C allow reconstruction of the $\delta^{18}O$ of the water ($\delta^{18}O_{water}$) in which they precipitate.** Solid grey
curves represent the calculated $\Delta_{47}$ and $\delta^{18}O_{carbonate}$ compositions of carbonates precipitated at oxygen
isotope equilibrium from water with fixed $\delta^{18}O_{water}$ values (indicated on each curve) and variable
temperatures. Horizontal dashed grey lines are calculated for fixed temperatures and variable $\delta^{18}O_{water}$.
The average $\delta^{18}O_{water}$ value of -6.6±0.4‰ measured in our experiments is reported using the thick solid
grey curve. The solid black curve was obtained using the $\delta^{18}O_{water}$ calculated with Eq. 1 (-8.0±2.8‰)
with its associated errors (dashed black curves).






$\Delta_{47}$ and $\delta^{18}O_{carbonate}$ values at equilibrium with: —— Calculated $\delta^{18}O_{water}$ - - - Measured $\delta^{18}O_{water}$



**Figure 4 | $\Delta_{47}$ and $\delta^{18}O_{carbonate}$ relationship to precipitation water $\delta^{18}O_{water}$ for other solid carbonates presenting oxygen isotope disequilibria.** In (**a**) to (**c**) black data series (this study, performed at 30°C) shows how kinetic oxygen isotope fractionation in the DIC prior to carbonate precipitation can be mistaking for high temperature isotopic equilibrium. Similarly to Fig. 3, the solid curves were obtained using the $\delta^{18}O_{water}$ calculated with Eq. 1. (**a**) Abiotic carbonates from Tang et al., (2014) illustrating the effect of $CO_2$ hydroxylation on $\Delta_{47}$ and $\delta^{18}O_{carbonate}$ values (various pH plotted with different colors, various temperatures plotted with different symbols). (**b**) Abiotic carbonates from Staudigel and Swart, (2019) illustrating the effect of $CO_2$ degassing and DIC oxygen isotope equilibration with water on $\Delta_{47}$ and $\delta^{18}O_{carbonate}$ values. $\Delta\delta^{13}C_{carbonate}$ stands for the difference between the $\delta^{13}C$ value measured in carbonates and the final $\delta^{13}C$ of the data series at the end of equilibration (various $\Delta\delta^{13}C$ ranges plotted with different colors, various temperatures plotted with different symbols). (**c**) Coccolithophorid *E. huxleyi* grown at 7, 10, 15, 20 and 25°C from Katz et al., (2017) showing how coccoliths with equilibrium $\Delta_{47}$ values record the equilibrium $\delta^{18}O_{water}$ of their body water, which differs from that of the culture medium (*i.e.* artificial seawaters A, B, and C plotted with different colors).



686

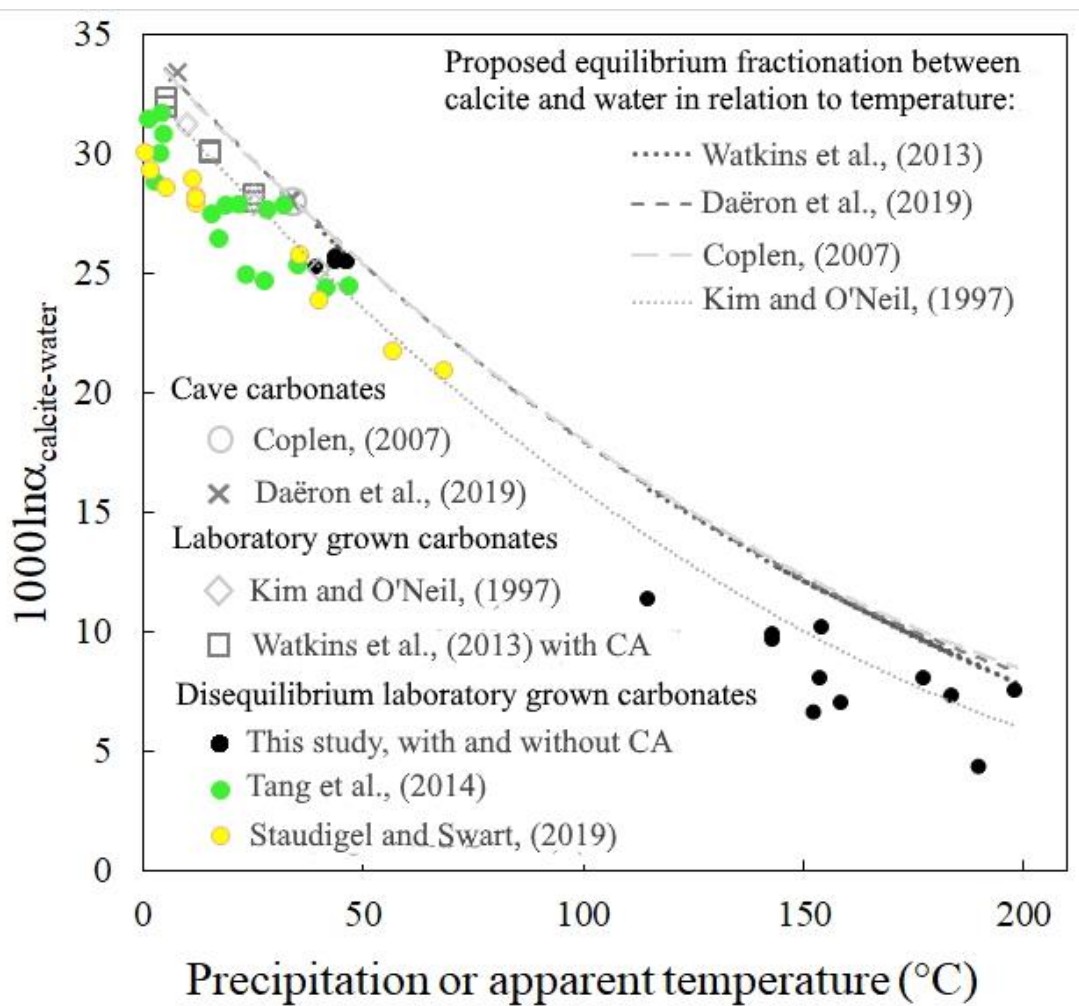

687

**Figure 5 | The relation to temperature of equilibrium oxygen isotope fractionation factor between**
**calcium carbonate and water ($1000\ln\alpha_{calcite-water}$) appears to be retrievable from solid carbonates**
**(mainly calcites) in strong clumped and oxygen isotope disequilibrium** such as our microbial
carbonates (black dots, precipitated at 30°C) and two additional data series of laboratory grown
carbonates showing disequilibrium fractionation (Tang et al., 2014; Staudigel and Swart, 2019) (green
and yellow dots, respectively). The data points affected by $CO_2$ hydroxylation (Tang et al., 2014) or $CO_2$
degassing (Staudigel and Swart, 2019) (see Fig.4) are not included. Grey symbols correspond to cave
carbonates precipitated at or near equilibrium (Coplen, 2007; Daëron et al., 2019) or laboratory



experiments (Kim and O'Neil, 1997; Watkins et al., 2013). Those grey data series are usually considered
as representative of the equilibrium fractionation factor between calcium carbonate and water whose
relations to temperature, extrapolated at high temperature, are illustrated by the different dashed curves.
Plotted temperatures corresponds to precipitation temperatures except for disequilibrium carbonates for
which apparent temperatures have been calculated based on $\Delta_{47}$ values. Errors are included in the
symbol size.