# Peer review of "Oxygen isotope composition of waters recorded in carbonates in strong clumped and oxygen"

_Biogeosciences, 2019_

## Referee Comment (RC1) · William Defliese (Referee) · 2 Jan 2020

This paper by Thaler et al. is very well written, and I think would be of interest to a large number of people in the scientific community. The finding that disequilibrium precipitation can potentially be used to track 'true' equilibrium is novel, and potentially very exciting. I find the paper well written, and I have few comments or questions that the authors do not answer. My only general comment is that I caution the authors to not declare one set of d18O calibrations/measurement to be representative of equilibrium versus another as of this point. I (and probably many others in the community) am VERY sceptical of assertions that a single datapoint represents isotopic equilibrium for

d18O and that all others do not. Overall, this is a good paper and I'm interested to see where this goes next.

Abstract: Good

Introduction: Good

Section 2.1: I'd like to see some statement about the mineralogy of the carbonates. It looks like that is reported to some extent in the supplement, but I think a sentence on mineralogy of your precipitates should go in the methods.

Section 3.1

Lines 171 and 172: Maybe mention in the text the equilibrium value and measured value, so that readers do not think -0.27 per mill is the measured value?

Section 3.2

Line 197: Do you know the isotopic composition of the urea? This should be easily testable. I realize it doesn't impact the results of this study, but would be nice to know! Section 3.3

Line 239: "using the calibration..."

Section 3.4

Lines 278-280: This is true, with the important exception of carbonates in which d13C is also out of equilibrium, which is discussed later in section 3.4.

Line 297: "be explained solely by temperature..."

Line 340: "form in caves from CO2..."

Section 3.6: I'm not sure you can really make any statements here about better matching the Coplen (2007) and Watkins et al. (2013) data at low temperature, as figure 5 does not show error bars. When the errors are plotted (particularly important for the D47-based temperatures), it looks to me like your data overlaps both d18O calibration

relationships.

Figure comments:

Figure 1: I'm confused by the x-axis. Why is CaCO3 being presented in units of millimolar, which is appropriate for a dissolved solution? A solid should be presented in units of mass, i.e. milligrams, or alternatively as moles precipitated.

Figure 2: Same comment as per figure 1.

Figure 5: The trendline symbols for Watkins (2013) and Kim and O'Neil (1997) are too similar, and difficult to distinguish. Also I'm a bit confused by the Watkins data, why are the symbols for Watkins on the same line as Kim and O'Neil yet the trendline lies above? Additionally, if it doesn't clutter up the plot too much it would be good to see the error bars on the carbonate datapoints.

―――――――――――――――――――

---

## Referee Comment (RC2) · Stefano Bernasconi (Referee) · 15 Jan 2020

This paper proposes that the deviation from equilibrium for clumped isotopes and oxygen isotopes is closely related by a constant relationship, and that due to this relationship the isotopic composition of the delta 18O value of paleowaters can be reconstructed even from carbonates precipitated out of equilibrium. This is an interesting concept, but it to be reinforced by using additional calibration curves for clumped and oxygen isotopes (see details below) before the conclusions can be considered robust. In particular the Bonifacie et al.2017 curve is not calculated with the "Brand" parameters whereas the data presented here are, thus there is the possibility of an offset

(see comment below) thus alternatives have to be considered. Similarly, there are other curves for calcite-water oxygen isotope fractionation that are probably closer to equilibrium (see below) and these alternatives should also be tested to evaluate the robustness of the conclusions. In addition, there are also other datasets available in the literature on dolomite that can be used to support the conclusions of this paper, and these should be included as well.

As the paper is not very long, it would be better to incorporate the supplementary information into the main manuscript, this would make the paper more easily readable as all the important information is in one document.

Detailed comments:

Line 29: The delta 18O of carbonate not "the delta 18O composition of carbonate"

Line 32 Abundance not abundancy

Line 36-37 these two papers report disequilibruium but these are extreme cases, and not so common,. In particular other occurrences of Methane seep carbonates have been shown to precipitate in equilibrium (see Zhang et al. EPSL 512, 207-213).

Line 38 change to "in some biogenic carbonates" Disequilibrium is found in corals and possibly in brachiopods, but other widespread carbonates do not show disequilibrium (foraminifera, Peral et al. 2019 for example are in equilibrium). As the sentence is formulated here it seems to indicate that disequilibrium is dominant in biogenic carbonates, which is not the case. This may suggest that the clumped isotope thermometer is seldom useful. Please correct.

Line 121 change to ". . .measured ratios of the sample CO2"

Line 128 "Equilibrium Scale " not "Equilibrated Scale"

Line 150 and supplementary information: The Bonifacie et al. 2017 is not calculated with the Brand parameters, thus it should not be used to calculate temperatures of samples whose D47 is calculated using the Brand parameters, as there could be an offset. For example the Kele et al. 2015 calibration before recalculation was indistinguishable from the Bonifacie et al. 2017 common calibration (it was also part of it). However, upon recalculation by Bernasconi et al. 2018; Geochem. Geophys. Geosys.), the intercept has changed by -38 ppm, thus the authors should be careful in establishing "equilibrium" with an equation that is not based on the Brand parameters. The temperature difference calculated with the recalculated Kele et al.2015 instead of the Bonifacie et al. 2017 is up to 25 degrees lower for the samples with high disequilibrium. The samples produced with the carbonic anhydrase with the Bernasconi et al. 2018 calibration give yield temperatures of 35 to 40°C about 5°C closer to equilibrium than using the Bonifacie et al. 2017 calibration. Does this change the interpretation of the trends? This should be taken in consideration and discussed. Many of the calibrations used in the Bonifacie calibration were indeed recalculated by Petersen et al. 2019; G3) and the values of some of the datasets changed significantly. It is difficult to evaluate what difference it would make to the Bonifacie curve, but this should be checked.

Due to these uncertainties, the inferences on difference to equilibrium could be biased and the calculated water compositions as well. I suggest that the temperatures should be recalculated using for example the recalculated Kele et al. 2015 and see how the interpretations may change.

The same discussion is valid also for the Kim and O'Neil calibration, does that really represent equilibrium? More and more evidence is that it is not (see discussion in Daeron et al. 2019). How would the interpretations in this paper change if another oxygen isotope calibration would be used ot calculate the oxygen isotope temperature? For example the Daeron et al. 2019 or O'Neil et al. (1969)? This point should be tested and possible implications discussed also in the discussion section.

Even if this is given in detail in Thaler et al. 2017, it would be useful for this paper to put a figure with the reaction pathway for ureolysis. It would be useful to have a formula which shows where the oxygen in the carbonate molecule comes from.

Line 201: its better to talk about depleted in 18O rather than enriched in 16O which is the more abundant isotope

Lines 211 -2015 Schmid 2011 (ETH dissertation, https://doi.org/10.3929/ethz-a-006551449) also reported some analyses of carbonates produced by ureolytic bacteria in strong disequilibrium, and data from carbonates produced by direct hydroxylation at high pH showed the effect of hydroxylation on the clumped isotopes of carbonates. The high D47 of the carbonates formed from directly hydroxylated CO2 can also be due to the direct inheritance of the isotopic composition of the CO2 as at high pH the equilibration time with water is very long, longer than the precipitation rate of carbonate.

Lines 268 The Kim & O'Neil calibration is based on samples precipitated to temperatures between 10 and 40 thus not the best one for higher temperatures. Simply the fact that is the most used does not mean that it it's the best one to use.

Lines 285ff It is not true that the Tang and Staudigel data are the only published data reporting, water, delta 18O Water and delta 18 calcite. The authors should include the Kele et al. (2015) data with the recalculated D47 in Bernasconi et al. 2018 in their analysis as they also have all the necessary data 18 o carb 18O water and clumped isotopes to test the validity of their hypothesis. Additional datasets that would be interesting to test would be the Dolomite data of Bonifacie et al. 2017 (op. cit) and the dolomite data of Müller et al. 2019 (Chem. Geol., 525, 1-17) I think this would make the message of this paper much more robust.

---

## Author Comment (AC1) · 17 Feb 2020

**Note:** *The comments of Dr Defliese are in bold italics,* the modification in the text are in red

- *This paper by Thaler et al. is very well written, and I think would be of interest to a large number of people in the scientific community. The finding that disequilibrium precipitation can potentially be used to track 'true' equilibrium is novel, and potentially very exciting. I find the paper well written, and I have few comments or questions that the authors do not answer. My only general comment is that I caution the authors to not declare one set of d18O calibrations/measurement to be representative of equilibrium versus another as of this point. I (and probably many others in the community) am VERY sceptical of assertions that a single datapoint represents isotopic equilibrium for d18O and that all others do not. Overall, this is a good paper and I'm interested to see where this goes next.*

We would like to thank Dr William Defliese for accepting to review our work and for his encouraging comments. We are pleased that he found our article well written and our scientific finding to be novel and promising. Dr Defliese raised some comments to which we answered both in this letter and by modifying the text of the manuscript. We hope it improved the clarity of the manuscript.

- *Abstract: Good Introduction: Good*

- *My only general comment is that I caution the authors to not declare one set of d18O calibrations/measurement to be representative of equilibrium versus another as of this point. I (and probably many others in the community) am VERY sceptical of assertions that a single datapoint represents isotopic equilibrium for d18O and that all others do not.*

- *Section 3.6: I'm not sure you can really make any statements here about better matching the Coplen (2007) and Watkins et al. (2013) data at low temperature, as figure 5 does not show error bars. When the errors are plotted (particularly important for the D47-based temperatures), it looks to me like your data overlaps both d18O calibration relationship.*

-We fully agree with Dr Defliese on this point. We regret that our manuscript could give the impression that we chose some calibrations as establishing the "true" equilibrium value. In order to deliver a much clearer message on this point, we modified the manuscript as follows:

Line 457: From our results, due to our experimental conditions and the associated error in our dataset, it is not possible and not our intention to argue in favor of one of these calibrations; This however shows how crucial it is to improve knowledge on the equilibrium $1000\ln\alpha_{\text{carbonate-water}}$ at both high and low temperatures in order to improve the accuracy and precision of our new proxy for reconstructing the $\delta^{18}O_{\text{water}}$ from which carbonates, even disequilibrium ones, precipitated.

-Additionnally, the $\Delta 47$ based temperature errors are now plotted on the mentioned figure

- *Section 2.1: I'd like to see some statement about the mineralogy of the carbonates. It looks like that is reported to some extent in the supplement, but I think a sentence on mineralogy of your precipitates should go in the methods.*

We have added a new figure 1 (formerly in the supplementary material) presenting a scanning electron microscopy (SEM) picture of the carbonates and the following lines in section 2.1:

Line 100: The major part of the carbonates precipitated in this study was composed of calcite (Thaler et al., 2017) with minor amounts of aragonite (1 to 4%), vaterite (2 to 4%) and magnesian calcite with low Mg content $(Mg_{0.064},Ca_{0.936})CO_3$ (up to 8%) (Thaler et al., 2017).

- *Section 3.1 Lines 171 and 172: Maybe mention in the text the equilibrium value and measured value, so that readers do not think -0.27 per mill is the measured value?*

We modified the text as follows:

Line 206: The $\Delta_{47}$ offset to equilibrium starts down to -0.270‰ (the largest $\Delta_{47}$ offset ever measured in solid carbonates) and the offset to equilibrium reaches -24.7‰ for $\delta^{18}O_{carbonate}$.

- *Section 3.2 Line 197: Do you know the isotopic composition of the urea? This should be easily testable. I realize it doesn't impact the results of this study, but would be nice to know!*

This is an interesting question to which we had to answer in Thaler et al., 2017. In this study, we measured the $\delta^{13}C$ value of urea, but we still do not know neither its $\delta^{18}O$ value nor its clumped isotope ratio. But even if we could have measured them, the system stays under constrained for the following reasons:

- If urea conversion into dissolved inorganic carbon (DIC) is total, its $\delta^{13}C$ is transferred as is to the DIC (despite the ~12 permil enzymatic fractionation associated with that conversion step), and then to the carbonates if all the DIC precipitates. It is not the case for $\delta^{18}O$ and thus for the clumped isotope ratio, due to the ureolysis reaction mechanism, which is a double hydrolysis (please see below).
- 2/3 of the oxygen atoms in the DIC produced by ureolysis come from water (with an unknown fractionation), and 1/3 from urea (with a fractionation that can be considered = 1 if all of the urea converts into DIC). Thus what matters the most is not the initial isotopic composition of urea but rather the isotopic value of the enzymatic reaction product. In our case, this product can be $CO_2$ which is then hydrated and/or hydroxylated, a step that adds an additional fractionation step. We unfortunately cannot measure the $\delta^{18}O$ nor the clumped composition of the $CO_2$ that just got produced.

We however have calculated in Thaler et al.,2017 that:

"With equilibrium initial conditions [we hypothesized here that $CO_2$ is beeing produced with an isotopic composition in equilibrium with water], CO2(aq) hydration and hydroxylation can explain a 13‰ offset from equilibrium for the $1000ln\alpha HCO_3^--H_2O$ value whereas the initial offset from equilibrium observed in solid carbonates was -24.7‰ in the experiment with CA. This demonstrates that CO2 cannot be in oxygen isotope equilibrium with water when it is generated by ureolysis."

Note that in response to one of Pr Bernasconi's comment, we now describe in the manuscript the ureolysis mechanism:

line 82: Ureolysis consists in 2 successive hydrolysis steps: (i) the hydrolysis of urea into ammonia ($NH_3$) and carbamate ($H_2N$-COOH) ($H_2N - CO - NH_2 + H_2O \rightarrow NH_3 + H_2N - COOH$), which is catalyzed by urease and is the rate limiting step, and (ii) the rapid and spontaneous hydrolysis of carbamate into ammonia and $CO_{2(aq)}$ ($H_2N - COOH + H_2O \rightarrow NH_3 + CO_{2(aq)} + H_2O$) (Krebs and Roughton, 1948; Matsuzaki et al., 2013) or $H_2CO_3$ ($H_2N - COOH + H_2O \rightarrow NH_3 + H_2CO_3$) (Mobley and Hausinger, 1989; Krajewska, 2009).

- *Section 3.3 Line 239: "using the calibration..."*

Correction made

- *Section 3.4 Lines 278-280: This is true, with the important exception of carbonates in which d13C is also out of equilibrium, which is discussed later in section 3.4.*

That is very true and needed to be highlighted in that sentence. We modified it as follows:

Line 309: "Note that such precision in $\delta^{18}O_{water}$ values found in disequilibrium carbonates is remarkable considering that even for carbonates at isotopic equilibrium for both $\delta^{13}C$ and $\delta^{18}O$, $\delta^{18}O_{water}$ can only be retrieved from paired $\Delta_{47}$ and $\delta^{18}O_{carbonate}$ values with a precision of $\pm1$‰ at best (see Supplementary information)."

- ***Line 297: "be explained solely by temperature..."***

Correction made

- ***Line 340: "form in caves from CO2..."***

Correction made

- ***Figure 1: I'm confused by the x-axis. Why is CaCO3 being presented in units of millimolar, which is appropriate for a dissolved solution? A solid should be presented in units of mass, i.e. milligrams, or alternatively as moles precipitated. Figure 2: Same comment as per figure 1.***

This mM unit was chosen to allow comparison with the experiment presented in Thaler et al., 2017. Having the amount of carbonates in mM unit also permits to directly compare it to the amount of DIC and dissolved inorganic nitrogen (DIN) produced in the solution. This latter points out that not all of the DIC precipitates, which has implication for the expression of several fractionation factors.

The comparison between the amount of carbonate formed and the DIN is now available in the new Figure 1, that was previously in the supplementary material of the manuscript

Although we agree mM is an odd unit to use for solids, we see some interest in keeping it that way, and would prefer not to change it.

- ***Figure 5: The trendline symbols for Watkins (2013) and Kim and O'Neil (1997) are too similar, and difficult to distinguish.***

We modified the trendline used.

- ***Also I'm a bit confused by the Watkins data, why are the symbols for Watkins on the same line as Kim and O'Neil yet the trendline lies above?***

We agree this may be confusing, but that is what Watkins et al., 2013 and then Watkins and Hunt 2015 propose.

Below is displayed Figure 5 from Watkins et al., 2013 and its legend, where their data are represented along with cave samples from Coplen, 2007.

[Figure]

**Fig. 5.** Relationship between the observed oxygen isotope composition of calcite and temperature. The grey shaded area represents the range of proposed equilibrium values. The dashed lines show the equilibrium oxygen isotope composition of $HCO_3^-$ and $CO_3^{2-}$ relative to water. The solid line is parallel to the equilibrium $CO_3^{2-}$–water line to illustrate that nearly all of the temperature-dependence of oxygen isotope fractionation between calcite and water can be attributed to the temperature dependence of oxygen isotope fractionation between $CO_3^{2-}$ (or $HCO_3^-$) and water. The temperature-dependence observed in the literature and in our catalyzed experiments does not necessarily imply oxygen isotope equilibrium between calcite and water. The data point from a natural cave calcite may represent thermodynamic equilibrium; it precipitated several orders of magnitude more slowly than experimental calcites (see Fig. 6).

The experiment catalyzed with CA presented in Watkins et al., 2013 falls on the solid line in their Figure 5. Even though it is not mentioned in the legend, it appears that this solid line is quite hard to distinguish from the line corresponding to Kim and O'Neil (1997) equation (i.e., 1000lna=18.03*1000/(T°C+273.15)-32.42).

As an illustration, at 0°C (where all curves intercept the Y-axis) the $1000ln\alpha_{calcite-water}$ value calculated with Kim and O'Neil (1997) equation (in other words $\Delta^{18}O_{calcite-water}$) equals 33.6, and at 35°C, $1000ln\alpha_{calcite-water}$ equals 26.1.

Accordingly, Watkins et al., 2013's catalyzed experiment is indeed on the same trendline as Kim and O'Neil, 1997, below the cave carbonates of Coplen, 2007.

However, we have also reproduced, again below, Figure 6 from Watkins et al., 2013 with its legend, where the model calibration equation ($\Delta^{18}Oeq_{calcite-water}$=17747/(T°C+273,15)-29.777) is given:

At 0°C, $\Delta^{18}Oeq_{calcite-water}$ = 35.2 and at 35°C, $\Delta^{18}Oeq_{calcite-water}$ = 27.8 (On Watkins et al., 2013's Figure 5, these two points are above the solid line).

So indeed, Watkins et al., 2013's model follows Coplen (2007) trendline even though the CA-catalyzed experiment datapoints fall on Kim and O'Neil (1997) curve. The catalyzed experiment datapoints fall on Watkins et al (2013) model trend line when represented against the precipitation rate (logR) that is incorporated in the calculation that produced Figure 6 lines. However, no $1000ln\alpha$ equation corrected for logR value is given in the figure.

[Figure]

**Fig. 6.** $\Delta^{18}O_{c-w}$ versus precipitation rate $R$. The model curves describe the rate-dependence on $\Delta^{18}O_{c-w}$ while accommodating the equilibrium $\Delta^{18}O_{c-w}$ at 33.7 °C and pH = 7.4 inferred from natural cave calcite (Coplen, 2007). The slowly grown cave calcites from Tremaine et al. (2011) formed at about 20 °C and pH≈7.8. In the model, the temperature dependence arises solely from temperature-dependent partitioning of oxygen isotopes between DIC species and water. The pH dependence arises from the difference in $HCO_3^-$ versus $CO_3^{2-}$ participating in calcite growth as a function of growth rate. The data of Dietzel et al. (2009) are isotopically light, which may be a consequence of isotopic disequilibrium among DIC species in the bulk solution from which calcite is precipitated. Such a disequilibrium effect is not incorporated into the model. An important insight from the agreement between model and data is that direct measurement of equilibrium $\Delta^{18}O_{c-w}$ by conventional methods may require experiments that last years to decades.

As you might wonder, Coplen (2007) exact equation is:

$$\ln \alpha_{calcite-water} = 17.4(1000/T) - 28.6 \qquad (3)$$

To confirm previous statements (that is a bit puzzling), below is Table 1 from Watkins and Hunt, 2015 with the calcite-water equilibrium equation used. Again Coplen, 2007 and Watkins et al., 2013 are undistinguishable:

**Table 1**

Compilation of equilibrium fractionation factors ($\alpha_{i-j}^{eq}$ unless otherwise noted) for carbon and oxygen isotopes in aqueous solution. These equations are presented graphically in Fig. 1 (note: $\Delta = 1000 \ln \alpha$).

| Compounds | Equation | $\alpha$ (25 °C) | References |
|---|---|---|---|
| **Carbon isotopes** | | | |
| $CO_2(g) - HCO_3^-$ | $-9.483/T_K + 1.0239$ | 0.9921 | Mook et al. (1974) as in Mook (1986) |
| $CO_2(aq) - HCO_3^-$ | $-9.866/T_K + 1.0241$ | 0.9910 | Vogel et al. (1970) as in Mook (1986) |
| $CO_3^{2-} - HCO_3^-$ | $-0.867/T_K + 1.0025$ | 0.9996 | Turner (1982) as in Mook (1986) |
| $CO_2(g) - $ calcite | $\Delta = -2.988 \cdot 10^6 / T_K^2$ | 0.9897 | Bottinga (1968) |
| | $\quad + 7.666 \cdot 10^3 / T_K - 2.461$ | | |
| | | | |
| **Oxygen isotopes** | | | |
| $CO_2(g) - H_2O$ | $17.611/T_K + 0.9821$ | 1.0412 | Zeebe (2007) |
| $CO_2(aq) - H_2O$ | $17.54/T_K + 0.9827$ | 1.0415 | Wang et al. (2013) |
| $HCO_3^- - H_2O$ | $17.76/T_K + 0.9725$ | 1.0321 | Wang et al. (2013) |
| $CO_3^{2-} - H_2O$ | $21.72/T_K + 0.9539$ | 1.0268 | Wang et al. (2013) |
| Calcite $- H_2O$ | $\Delta = 17747/T_K - 29.777$ | 1.0302 | Coplen (2007), Watkins et al. (2013) |

- *Additionally, if it doesn't clutter up the plot too much it would be good to see the error bars on the carbonate datapoints.*

-We chose to only mention the error on our data point in the legend, not only to not clutter up the plot, but also because no error is given in Kim and O'Neil, 1997, which makes error comparison between datasets complicated. We thus chose to not represent any error in the first place. However, please note that the symbol size is quite big and probably encompass the error (if it is of around 0.3 or 0.4 permil, which is plausible).

-The $\Delta_{47}$ based temperature errors are now plotted on the mentioned figure. The following sentence has also been added to the associated legend:

"X-axis errors for this study are included in the symbol size. The Y-axis error for all the reconstructed temperature is given in the figure."

**Bibliography**

Coplen, T. B.: Calibration of the calcite–water oxygen-isotope geothermometer at Devils Hole, Nevada,

   a natural laboratory. *Geochim. Cosmochim. Acta* **71**, 3948–3957,

   https://doi.org/10.1016/j.gca.2007.05.028, 2007.

Watkins, J. M., Nielsen, L. C., Ryerson, F. J. & DePaolo, D. J.: The influence of kinetics on the oxygen

   isotope composition of calcium carbonate. *Earth Planet. Sci. Lett.* **375**, 349–360,

   https://doi.org/10.1016/j.epsl.2013.05.054, 2013.

Watkins, J. M. & Hunt, J. D.: A process-based model for non-equilibrium clumped isotope effects in

   carbonates. *Earth Planet. Sci. Lett.* **432**, 152–165, https://doi.org/10.1016/j.epsl.2015.09.042, 2015.

Kim, S. T. & O'Neil, J. R.: Equilibrium and nonequilibrium oxygen isotope effects in synthetic

   carbonates. *Geochim. Cosmochim. Acta* **61**, 3461–3475, https://doi.org/10.1016/S0016-

   7037(97)00169-5, 1997.

---

## Author Comment (AC3) · 17 Feb 2020

*(**The comment of Pr Bernasconi are in bold italics,** the modification in the text are in red)*

> ***This paper proposes that the deviation from equilibrium for clumped isotopes and oxygen isotopes is closely related by a constant relationship, and that due to this relationship the isotopic composition of the delta 18O value of paleowaters can be reconstructed even from carbonates precipitated out of equilibrium. This is an interesting concept, but it to be reinforced by using additional calibration curves for clumped and oxygen isotopes (see details below) before the conclusions can be considered robust. In particular the Bonifacie et al.2017 curve is not calculated with the "Brand" parameters whereas the data presented here are, thus there is the possibility of an offset (see comment below) thus alternatives have to be considered. Similarly, there are other curves for calcite-water oxygen isotope fractionation that are probably closer to equilibrium (see below) and these alternatives should also be tested to evaluate the robustness of the conclusions. In addition, there are also other datasets available in the literature on dolomite that can be used to support the conclusions of this paper, and these should be included as well.***

We would like to thank Pr. Bernasconi for agreeing to evaluate our work. His review raises important points which we are pleased to clarify in this letter. Some points also required modification that figure in the the revised manuscript, inducing minor changes that however do not change our initial conclusions.

First of all, we would like to stress out that the goal of the paper is not to establish equilibrium but rather to suggest, for future studies, an original method to approach it better. Indeed, future well-designed experiments associated with accurate isotopic measurements should permit to better approach equilibrium equations. Since it is crucial that our objective here is clearly understood by readers (ie. that *our paper does not aim to establish equilibrium*), we have added the following sentence:

Line 457: "From our results, due to our experimental condition and the associated error in our dataset, it is not possible and not our intention to argue in favor of one of these calibrations. This however shows how crucial it is to improve knowledge on the equilibrium $1000\ln\alpha_{\text{carbonate-water}}$ at both high and low temperatures in order to improve the accuracy and precision of our new proxy for reconstructing the $\delta^{18}O_{\text{water}}$ from which carbonates, even disequilibrium ones, precipitated."

Second, we would like to stress out that, in the original submission of the manuscript, we did take into account several calibrations both for clumped and oxygen isotopes, arguing the choice of calibration both for $\Delta_{47}$ and $\delta^{18}O$ would not change our conclusions (e.g., Kelson 2017 versus Bonifacie 2017, former lines 150 to 152; or comparing our reconstructed $\delta^{18}O_{\text{water}}$ with several combination of $\delta^{18}O_{\text{water-calcite}}$ calibrations former lines 420 to 422, all of this text remains in the new version of the manuscricpt).

Detailed answer to this comment and to other specific comments about using a calibration calculated with Brand parameters or using dolomite calibrations are detailed underneath, after the more detailed comments made by S. Bernasconi.

- ***As the paper is not very long, it would be better to incorporate the supplementary information into the main manuscript, this would make the paper more easily readable as all the important information is in one document.***

We initially considered that the technical information necessary to the clumped isotope community scientist to evaluate and reproduce our analysis and calculation protocols are not of interest for the greater biogeosciences community and hence provided them in the supplementary material with a summary (paragraph 2.2. former lines 99 to 142) in the main text. We however decided to move the following items, of interest to the broader biogeosciences community, from the supplementary material to the main text following S. Bernasconi recommendation:

-the ureolysis mechanism, new lines 82 to 86
- Supplementary Figure 1, (now fig. 1) that helps the reader to appreciate how close to the cells precipitation is occurring
- Supplementary figure 2, (now Fig. 2) that permits to better understand how the microbial ureolysis triggers carbonate precipitation
- Supplementary text explaining the rationale for the choice of D47 calibration, new lines 168 to 193

- ***Line 29: The delta 18O of carbonate not "the delta 18O composition of carbonate"***

Correction made

- ***Line 32 Abundance not abundancy***

Correction made

- ***Line 36-37 these two papers report disequilibruium but these are extreme cases, and not so common,. In particular other occurrences of Methane seep carbonates have been shown to precipitate in equilibrium (see Zhang et al. EPSL 512, 207-213).***

We agree that strong disequilibrium is not common, however disequilibrium (strong enough to not be able to reconstruct correct precipitation temperature) is common. As highlighted recently by Daeron et al., 2019: "Most earth surface calcites precipitate out of isotopic equilibrium" which of course do not signify that no carbonates can precipitate at least close to equilibrium. However, the novelty of this article is specifically to evaluate the information that can be obtained from carbonates formed with an identifiable isotopic disequilibrium, which is why we focus on these datasets. Carbonates formed at isotopic equilibrium are already usable to reconstruct paleoclimates with traditional approaches, and are not the topic of our study.

- ***Line 38 change to "in some biogenic carbonates" Disequilibrium is found in corals and possibly in brachiopods, but other widespread carbonates do not show disequilibrium (foraminifera, Peral et al. 2019 for example are in equilibrium). As the sentence is formulated here it seems to indicate that disequilibrium is dominant in biogenic carbonates, which is not the case. This may suggest that the clumped isotope thermometer is seldom useful. Please correct.***

-The intention of the text (former) lines 36 to the end of the paragraph is to say that the vital effect issue is well known and widespread for $\delta^{18}O$ and increasingly identified for $\Delta_{47}$. The $\delta^{18}O$ vital effect can be strong in corals and brachiopods or coralline algae but it exists as well in coccoliths or foraminifera. We thus prefer to say that disequilibrium is common in biogenic carbonates. This does not prevent $\delta^{18}O$ and $\Delta_{47}$ tools to be useful as empirical calibrations taking vital effects into account allow temperature reconstructions.

That sentence has been added in the manuscript to prevent any reader to believe that vital effect identification invalidate the use of the isotopic thermometry.

Line 44 : Even though these vital effect identification do not prevent $\delta^{18}O$ and $\Delta_{47}$ tools to be powerful paleothermometer as empirical calibrations taking vital effects into account allow temperature reconstructions, it has become crucial to determine if the $\delta^{18}O$ and $\Delta_{47}$ disequilibria observed in carbonates as diverse as those found in coral reefs (Saenger et al., 2012), brachiopods (Bajnai et al., 2018), microbialites and methane seep carbonates (Loyd et al., 2016), along with speleothems (Affek et al., 2014) could be explained by oxygen-isotope disequilibria occurring in dissolved inorganic carbon (DIC) involved in carbonate precipitation.

-It is however true that vital effect is mostly studied on $\delta^{18}O$ data considering the difference in the temperature precision obtained with $\delta^{18}O$ reconstruction in comparison to $\Delta_{47}$ reconstruction, due to the sensibility to temperature of the parameters and/or to the precision of the measurements. We calculated below two examples, using a reasonable 0.2 permil error on a $\delta^{18}O$ measurements (many laboratory do better) and a good 0.008 permil error on a $\Delta_{47}$ measurement (many laboratory do not reach that precision) using two calibrations which use are proposed by the reviewers (the result would be similar with other calibrations):

| Used equation: $\Delta_{47}=0.0449*10^6/T^2+0.167$ (Bernasconi et al., 2018) | | | |
|---|---|---|---|
| | $\Delta_{47}$ - 0.008 | $\Delta_{47}$ real | $\Delta_{47}$ + 0.008 |
| Range of $\Delta_{47}$ comprised within the error: | 0.719 | 0.727 | 0.735 |
| recalculated T°C: | 12.0 | 10.0 | 8.0 |
| | error on T°C : +/- 2°C | | |

| Used equation: $\delta^{18}O_c-\delta^{18}O_w= 17747/T -29.777$ (Watkins et al., 2013) | | | |
|---|---|---|---|
| | $\delta^{18}O$ - 0.2 | $\delta^{18}O$ real | $\delta^{18}O$ + 0.2 |
| Range of $\delta^{18}O_c$ comprised within the error: | 32.7 | 32.9 | 33.1 |
| recalculated T°C (for a $\delta^{18}Ow = 0$): | 10.9 | 10.0 | 9.1 |
| | error on T°C : +/- 0.9°C | | |

The additional error associated to the equations parameters themselves is not taken into account.

Due to that difference in precision, vital effects can be masked in many species (except in coral, brachiopods or our microbial carbonates where clumped disequilibrium was big enough to be identified) in $\Delta_{47}$ studies, but it does not mean it is not there. We believe that these vital effects deserve being studied so that reconstructions get richer in information.

We acknowledge that Peral et al., 2018 show that with a precision of 0.008 they cannot distinguish species specific differences on the $\Delta_{47}$ in several species of foraminifera (while those species specific vital effects are known for $\delta^{18}O$). We also acknowledge that the calibration obtained on foraminifera agrees well with the kele 2015 calibration recalculated by Bernasconi et al. 2018. This indicates that paleoclimate reconstruction from $\Delta_{47}$ measurement on foraminifera, or from any "close to equilibrium" carbonates, measured with a 0.008 precision, will give similar temperature reconstruction precision. The fact that any vital effect cannot be identified today does not mean it will not when the precision gets better… especially when already identified on $\delta^{18}O$. We would like to add that in Peral et al., 2019 size specific vital effect on both $\delta^{18}O$ and $\Delta_{47}$ on the species G. inflate is indeed described.

- ***Line 121 change to ". . .measured ratios of the sample CO2"***

Correction made

- *Line 128 "Equilibrium Scale " not "Equilibrated Scale"*

Correction made

- *Line 150 and supplementary information: The Bonifacie et al. 2017 is not calculated with the Brand parameters, thus it should not be used to calculate temperatures of samples whose D47 is calculated using the Brand parameters, as there could be an offset. For example the Kele et al. 2015 calibration before recalculation was indistinguishable from the Bonifacie et al. 2017 common calibration (it was also part of it). However, upon recalculation by Bernasconi et al. 2018; Geochem. Geophys. Geosys.), the intercept has changed by -38 ppm, thus the authors should be careful in establishing "equilibrium" with an equation that is not based on the Brand parameters.*

We agree with S. Bernasconi on the fact that using a calibration not calculated with the Brand parameters to convert data themselves calculated with the Brand parameters is not ideal. However, this feature is not the only one to consider for the use of a calibration compatible with the acquired data. Then, as stressed out in the supplementary discussion of the initial submission (where we detailed the rationale for our choice of calibration) there are strong arguments why we have chosen (and still want) to use Bonifacie et al. 2017 calibration in our case, which we still believe is the most valid/compatible to compare to our data. We however acknowledge that this information might have been somehow hidden in supplementary info (as also mentionned by S. Bernasconi's above comment) in our initial submission, we thus moved some explanations/clarifications in the main text (as in supplementary information). See detailed comments and changes underneath

New line 168 (in the main text):
For the temperature (T) derived from the $\Delta_{47}$ data, we chose the calibration determined by Bonifacie et al., (2017) as it integrates a consequent number of data (n > 300), which statistical weight have been properly considered, and covers a wide temperature range (from 1 to 350°C), three characteristics that were recently shown by several teams as governing the precision on $\Delta_{47}$-T calibration equations (Bonifacie et al., 2017; Kelson et al., 2017; Fernandez et al., 2017). Importantly, this calibration covers the high apparent temperature ranges reported here (*i.e.,* low $\Delta_{47}$ values) allowing to avoid loss of precision/accuracy when extrapolating to temperature ranges that have not been experimentally investigated. Finally the Bonifacie et al. (2017) calibration has been checked independently with other methods (Mangenot et al., 2017, Dassié et al., 2018) on the range of temperatures (~30 to 96°C) where most of available calibrations are diverging and/or not well constrained. Indeed, these studies report excellent consistencies: i/ between T$\Delta_{47}$ and homogenization temperatures from fluid inclusion microthermometry (Mangenot et al., 2017), and *ii/* between the $\delta^{18}O_{water}$ values directly measured in fluid inclusions by cavity ring down spectroscopy and those calculated from combined T$\Delta_{47}$ and $\delta^{18}O_{carb}$ of the host-mineral (Dassié et al., 2018).

-For taking into account the brand parameter issue underlined here by S. Bernasconi, we still prefer to use the Kelson et al., 2017's calibration (see lines 159 to 161, present in our original submission) which is, as our data presented here, *directly* normalized with equilibrated gas standards only (that is the correction frame used by the whole community since 14 years) and with limited uncertainties because describing a large range in temperature, and acquired in a single lab (thus avoiding complications arising from comparing data from different laboratories, as shown in Petersen et al., 2019). Given the current consensus of using equilibrated gas standards, we believe that

comparing our data with Kele/Bernasconi 2018 (normalized to carbonate standards) would introduce much more confusion (and likely bias – see supplementary Table S4) than the comparison we currently propose.

That being said, it is noteworthy that significant improvements are currently undertaken by the clumped isotope community to minimize bias between laboratories by using carbonate standards to normalize data instead of equilibrated gases (S. Bernasconi is leading this INTERCARB project with 4 other clumped isotope researchers including M. Bonifacie co-author of this paper). The project has gathered more than 2000 measurements on the same standards from 26 different laboratories. But the results have only been communicated as internal reports to participants and are thus not yet public before their publications.

Interestingly, the about 38 ppm difference mentioned here by S. Bernasconi between the intercepts from Bonifacie et al. 2017 and Kele/bernasconi2018 calibrations (NB slopes are similar) is similar to the average offset found between IPGP and ETH labs on the four standards ETH1, ETH2, ETH3, ETH4 analyzed in both labs (see Table below, also now added as supplementary Table S4) and used to normalize the Kele/Bernasconi calibration.

To avoid confusion of readers, and because carbonate standards will likely be increasingly (if not exclusively) used in the future $\Delta_{47}$ measurements, we decided to add in the revised documents:

New line 181 (in main text):

Thought we recognize that the normalization to carbonate standards presented in Bernasconi et al. (2018) might become commonly used by the community in the future (ie. with the on-going inter-comparison Intercarb project), we prefer not to use this correction frame here because not enough of the four carbonate standards proposed by Bernasconi et al., 2018 were run together with our samples (n= 14 run in total of ETH1, ETH2, ETH3, ETH4 standards; Table S5), and such normalization method will then introduce larger uncertainty than the normalization we performed with the large number of equilibrated gases ran daily together with our unknowns (n= 104 equilibrated gas; Table S5 — Note also 33 secondary carbonate standards 102-GC-AZ01 and IPGP-Cararra, also ran in other IPGP studies and some other laboratories). Also remarkably, $\Delta_{47}$ obtained here on the four ETH carbonate standards are all systematically higher than values reported in Bernasconi et al., 2018 (Table S4). Thought the reason of this positive offset is still unclear, it is noteworthy that positive offsets are also observed when compiling other recent published values (Table S4; Daeron et al. 2016; Schauer et al., 2016; Fiebig et al., 2019; ).

With Table S4 (in supplementary material)

Because carbonate standards will likely be increasingly used in the future for normalizing $\Delta_{47}$ measurements, we here provide inter-laboratory comparison of $\Delta_{47}$ values obtained on the four standards provided by S. Bernasconi (ETH1, ETH2, ETH3, ETH4) as supplementary Table S4. This should also allow future use of our dataset.

| | Bernasconi et al. 2018 | | This study | | | | Fiebig et al. 2019 | | | | Daeron et al., 2016 | | | | Schauer et al., 2016 | | | |
|---|---|---|---|---|---|---|---|---|---|---|---|---|---|---|---|---|---|---|
| | D47CDES25 ‰ | D47CDES90* ‰ | D47CDES90 ‰ | ±1SD ‰ | n | offset ‰ | D47CDES90 ‰ | ±1SD ‰ | n | offset ‰ | D47CDES90 ‰ | ±1SD ‰ | n | offset ‰ | D47CDES25 ‰ | D47CDES90* ‰ | n | offset ‰ |
| ETH1 | 0.258 | 0.176 | 0.230 | | 1 | -0.054 | 0.214 | 0.009 | 19 | -0.038 | 0.229 | 0.024 | 18 | -0.053 | 0.287 | 0.205 | 8 | -0.029 |
| ETH2 | 0.256 | 0.174 | 0.222 | 0.009 | 4 | -0.048 | 0.215 | 0.011 | 18 | -0.041 | 0.224 | 0.024 | 13 | -0.050 | 0.281 | 0.199 | 12 | -0.025 |
| ETH3 | 0.691 | 0.609 | 0.620 | 0.012 | 8 | -0.011 | 0.619 | 0.009 | 16 | -0.010 | | | | | | | | |
| ETH4 | 0.507 | 0.425 | 0.464 | | 1 | -0.039 | 0.457 | 0.009 | 11 | -0.032 | | | | | | | | |
| | | | average offset | | | -0.038 | | | | -0.030 | | | | -0.052 | | | | -0.027 |

"D47CDES25" and "D47CDES90 "are D47 values reported versus the carbon dioxide equilibrated scale in the 25°C and 90°C acid digestion reference frame, respectively (in ‰)

"D47CDES90*" from Bernasconi et al., 2018 and Shchauer et al., 2016 calculated using
the D*25-90°C of 0.082‰ from Defliese et al. 2015
"n" is the number of replocate
measurements considered
"Offset" is the offset between D47CDES90 values reported in Bernasconi
et al and thos of the considered studies

- ***The temperature difference calculated with the recalculated Kele et al.2015 instead of the Bonifacie et al. 2017 is up to 25 degrees lower for the samples with high disequilibrium. The samples produced with the carbonic anhydrase with the Bernasconi et al. 2018 calibration give yield temperatures of 35 to 40∘C about 5∘C closer to equilibrium than using the Bonifacie et al. 2017 calibration. Does this change the interpretation of the trends? This should be taken in consideration and discussed. Many of the calibrations used in the Bonifacie calibration were indeed recalculated by Petersen et al. 2019; G3) and the values of some of the datasets changed significantly. It is difficult to evaluate what difference it would make to the Bonifacie curve, but this should be checked. Due to these uncertainties, the inferences on difference to equilibrium could be biased and the calculated water compositions as well. I suggest that the temperatures should be recalculated using for example the recalculated Kele et al. 2015 and see how the interpretations may change.***

First, we would like to stress out that the Bonifacie et al. 2017 calibration was mainly based (considering the statistical weight of the respective data, and _not just the number of data_ as usually made before [for more details on the statistical weight of respective data see discussion page 274 of Bonifacie et al., 2017]) on $\Delta_{47}$ data that did not significantly changed with the recalculation with the Brand parameters. Notably, the Kele 2015 data (which changed a lot with the Brand parameters as mentioned by S. Bernasconi) were only contributing to 10% to the final calibration. Thus changing to the recalculated $\Delta_{47}$ Brand values do not impact significantly the published Bonifacie et al., calibration. Importantly, datasets statistically contributing more to the calculated equations (ie. Wacker et al and Henkes et al. contributed as high as 40%) did not show large changes with the recalculations in Petersen et al (see Figure 3 of Petersen et al., 2019).

Second, we would like to show here that using kele/bernasconi calibration, as requested by S. Bernasconi, do not change our results and interpretations as shown in underneath figures.

[Figure]

For some points there is a better match for other there is a bigger difference, in general the agreement on the temperature recalculated on the data series with CA is indeed closer to equilibrium with Bernasconi et al., 2018 calibration, which is not in itself better or worse as, as explained in Thaler et al., 2017, the carbonates precipitated with CA corresponds to minerals that precipitate all along the equilibration, they thus should be off equilibrium a little to account for the initial "seeds" that precipitate at the beginning of the equilibration from a DIC in strong disequilibrium, and we do not know how the $\delta^{13}C$ different composition of the "seeds" impact the final $\Delta_{47}$. For the experiment Without CA, using bernasconi's equation pushes our data even further away from Coplen/Watkins/Daeron calibration, which again is not a bad thing as long as we consider that the question of the "true" equilibrium relation is not settled yet and it is not our intention to settle it in this paper.

Anyway, the main interpretation of our data is that similar wrong temperature can be reconstructed from both d18o and $\Delta_{47}$ measurement, all the conclusions on the reconstruction of $\delta^{18}O_{water}$ comes from that point, and this does not change at the first order. It would change the isotopic composition of the calculated water, but theoretically less

than using Watkins+bonifacie instead of Bonifacie and Kim and O'Neil (which would have return 2‰ lower values as already stated in the text).

- ***The same discussion is valid also for the Kim and O'Neil calibration, does that really represent equilibrium? More and more evidence is that it is not (see discussion in Daeron et al. 2019). How would the interpretations in this paper change if another oxygen isotope calibration would be used ot calculate the oxygen isotope temperature? For example the Daeron et al. 2019 or O'Neil et al. (1969)? This point should be tested and possible implications discussed also in the discussion section.***
- ***Lines 268 The Kim & O'Neil calibration is based on samples precipitated to temperatures between 10 and 40 thus not the best one for higher temperatures. Simply the fact that is the most used does not mean that it it's the best one to use.***

Our objective is not here to determine whether one calibration represents equilibrium better than another but rather to evidence that our knowledge on the equilibrium value still need to be refined, and to propose an experimental method to improve our knowledge on that topic. We thus prefer not to discuss how far from equilibrium Kim and O'Neil calibration is in comparison to other. As shown above our conclusions do not change with the calibration choice.

However, it is correct to say that the value of the reconstructed $\delta^{18}O_{water}$ changes with the calibration used. This was already discussed in the original submission and still in the manuscript with examples of effects:

"Coplen, (2007) or Watkins et al., (2013) equations would have return 2‰ lower values (ca. -10±2‰ compared to -8±3‰ calculated with Kim and O'Neil (1997) equation). From our results, it is not possible and not our intention to argue in favor of one of these calibrations "

- ***Even if this is given in detail in Thaler et al. 2017, it would be useful for this paper to put a figure with the reaction pathway for ureolysis. It would be useful to have a formula which shows where the oxygen in the carbonate molecule comes from.***

We have added the reaction mechanism in the method part:

line 82: Ureolysis corresponds to two hydrolysis : (i) the hydrolysis of urea into ammonia ($NH_3$) and carbamate ($H_2N$-COOH) ($H_2N - CO - NH_2 + H_2O \rightarrow NH_3 + H_2N - COOH$), which is catalyzed by urease and is rate limiting, and (ii) the rapid and spontaneous hydrolysis of carbamate into ammonia and $CO_{2(aq)}$ ($H_2N - COOH + H_2O \rightarrow NH_3 + CO_{2(aq)} + H_2O$) (Krebs and Roughton, 1948; Matsuzaki et al., 2013) or into $H_2CO_3$ ($H_2N - COOH + H_2O \rightarrow NH_3 + H_2CO_3$) (Mobley and Hausinger, 1989; Krajewska, 2009).

- ***Line 201: its better to talk about depleted in 18O rather than enriched in 16O which is the more abundant isotope***

Correction made

- ***Lines 211 -2015 Schmid 2011 (ETH dissertation, https://doi.org/10.3929/ethz-a006551449) also reported some analyses of carbonates produced by ureolytic bacteria in strong disequilibrium, and data from carbonates produced by direct hydroxylation at high pH showed the effect of hydroxylation on the clumped isotopes of carbonates. The high D47 of the carbonates formed from directly hydroxylated CO2***

*can also be due to the direct inheritance of the isotopic composition of the CO2 as at high pH the equilibration time with water is very long, longer than the precipitation rate of carbonate.*

We agree with that interpretation. This specific mechanism was developed in the original submission of the manuscript inside the paragraph on the effect of $CO_2$ hydroxylation/hydration on $\Delta_{47}$ values as illustrated by Tang et al abiotic experiment (still in the manuscript lines 342 to 358 and Figure 6).

- *Lines 285 It is not true that the Tang and Staudigel data are the only published data reporting, water, delta 18O Water and delta 18 calcite. The authors should include the Kele et al. (2015) data with the recalculated D47 in Bernasconi et al. 2018 in their analysis as they also have all the necessary data 18 o carb 18O water and clumped isotopes to test the validity of their hypothesis.*

Agreed. We clarified the text. We wanted to say that these are the only published data with disequilibrium carbonates that present a complete set of data. Equilibrium data series indeed often present a complete set of data. However it is not the focus of our study to determine the actual value of equilibrium which is why we do not present many "equilibrium" data series, only the most "famous" that scientist may already well know and may already have compared their data to. We corrected the sentence:
Line 317" These studies were chosen to further evaluate the relevancy of our $\delta^{18}O_{carbonate}$- $\Delta_{47}$ correlation because they are the only published dataset reporting full sets of *measured (rather than calculated)* $\delta^{18}O_{water}$, $\delta^{18}O_{carbonate}$ and $\Delta_{47}$ values, with *one or both proxies showing disequilibrium*, together with precipitation temperatures."

- *Additional datasets that would be interesting to test would be the Dolomite data of Bonifacie et al. 2017 (op. cit) and the dolomite data of Müller et al. 2019 (Chem. Geol., 525, 1-17) I think this would make the message of this paper much more robust*

We disagree with this proposition that we believe is beyond the scope of our study.  Plus, dolomites have a different $\delta^{18}O$ fractionation factor to water than calcite, and our samples (and the $\delta^{18}O$ calibrations used) are mostly calcites.

*Bibliography*

Daëron M., Blamart D., Peral M. and Affek H. P. (2016) Absolute isotopic abundance ratios and the

accuracy of D47 measurements.Chem. Geol. 442, 83–96.

Daëron, M., Drysdale, R. N., Peral, M., Huyghe, D., Blamart, D., Coplen, T. B., Lartaud, F. &

Zanchetta, G.: Most Earth-surface calcites precipitate out of isotopic equilibrium. *Nat. Commun.* **10**,

429, https://doi.org/10.1038/s41467-019-08336-5, 2019.

Dassié E. Genty, D., Noret, A., Mangenot, X., Massault, M., Lebas, N., Duhamel, M., Bonifacie, M., Gasparrini, M., Minster B., and Michelot, J.L. (2018) A newly designed analytical line to examine the reproducibility of fluid inclusion isotopic compositions in small carbonate samples. Geochemistry, Geophysics, Geosystems, 19 (4) 1107-1122. DOI: 10.1002/2017GC007289

Bonifacie, M. Calmels, D., Eiler, J. M., Horita, J., Chaduteau, C., Vasconcelos, C., Agrinier, P., Katz, A., Passey, B. H., Ferry, J. M., & Bourrand, J. J..: Calibration of the dolomite clumped isotope thermometer from 25 to 350° C, and implications for a universal calibration for all (Ca, Mg, Fe)CO$_3$ carbonates. *Geochim. Cosmochim. Acta* **200**, 255–279, https://doi.org/10.1016/j.gca.2016.11.028, 2017.

Bernasconi, S. M., Müller, I. A., Bergmann, K. D., Breitenbach, S. F., Fernandez, A., Hodell, D. A., ... & Ziegler, M. (2018). Reducing uncertainties in carbonate clumped isotope analysis through consistent carbonate-based standardization. Geochemistry, Geophysics, Geosystems, 19(9), 2895-2914.

Kele, S., Breitenbach, S. F., Capezzuoli, E., Meckler, A. N., Ziegler, M., Millan, I. M., ... & Yan, H. (2015). Temperature dependence of oxygen-and clumped isotope fractionation in carbonates: a study of travertines and tufas in the 6–95 C temperature range. Geochimica et Cosmochimica Acta, 168, 172-192.

Kelson, J. R., Huntington, K. W., Schauer, A. J., Saenger, C. & Lechler, A. R.: Toward a universal carbonate clumped isotope calibration: Diverse synthesis and preparatory methods suggest a single temperature relationship. *Geochim. Cosmochim. Acta* **197**, 104–131, https://doi.org/10.1016/j.gca.2016.10.010, 2017.

Watkins, J. M., Nielsen, L. C., Ryerson, F. J. & DePaolo, D. J.: The influence of kinetics on the oxygen isotope composition of calcium carbonate. *Earth Planet. Sci. Lett.* **375**, 349–360, https://doi.org/10.1016/j.epsl.2013.05.054, 2013.

Kim, S. T. & O'Neil, J. R.: Equilibrium and nonequilibrium oxygen isotope effects in synthetic carbonates. *Geochim. Cosmochim. Acta* **61**, 3461–3475, https://doi.org/10.1016/S0016-7037(97)00169-5, 1997.

Peral, M., Daëron, M., Blamart, D., Bassinot, F., Dewilde, F., Smialkowski, N., ... & Michel, E. (2018). Updated calibration of the clumped isotope thermometer in planktonic and benthic foraminifera. Geochimica et Cosmochimica Acta, 239, 1-16.Fiebig, J., Bajnai, D., Löffler, N., Methner, K., Krsnik, E., Mulch, A., & Hofmann, S. (2019). Combined high-precision$\Delta$ 48 and$\Delta$ 47 analysis of carbonates. Chemical Geology, 522, 186-191.

Schauer A. J., Kelson J. R., Saenger C. and Huntington K. W. (2016) Choice of 17O correction affects clumped isotope (D47) values of CO2 measured with mass spectrometry. Rapid Commun. Mass Spectrom. 30, 2607–2616.

Petersen, S. V., Defliese, W. F., Saenger, C., Daëron, M., Huntington, K. W., John, C. M., ... & Olack, G. A. (2019). Effects of improved 17O correction on interlaboratory agreement in clumped isotope calibrations, estimates of mineral-specific offsets, and temperature dependence of acid digestion fractionation. Geochemistry, Geophysics, Geosystems, 20(7), 3495-3519.